# Environment Constrains Fitness Advantages of Division of Labor in Microbial Consortia Engineered for Metabolite Push or Pull Interactions

Ashley E. Beck,[a,d] Kathryn Pintar,[a*] Diana Schepens,[b§] Ashley Schrammeck,[a◇] Timothy Johnson,[a∞] Alissa Bleem,[a‡] Martina Du,[a] William R. Harcombe,[c] Hans C. Bernstein,[e] Jeffrey J. Heys,[a] Tomas Gedeon,[b] Ross P. Carlson[a]

[a]Department of Chemical and Biological Engineering, Montana State University, Bozeman, Montana, USA
[b]Department of Mathematics and Statistics, Montana State University, Bozeman, Montana, USA
[c]Department of Ecology, Evolution, and Behavior, University of Minnesota, St. Paul, Minnesota, USA
[d]Department of Biological and Environmental Sciences, Carroll College, Helena, Montana, USA
[e]Norwegian College of Fisheries Sciences & The Arctic Centre for Sustainable Energy, UiT–The Arctic University of Norway, Tromsø, Norway

**ABSTRACT** Fitness benefits from division of labor are well documented in microbial consortia, but the dependency of the benefits on environmental context is poorly understood. Two synthetic *Escherichia coli* consortia were built to test the relationships between exchanged organic acid, local environment, and opportunity costs of different metabolic strategies. Opportunity costs quantify benefits not realized due to selecting one phenotype over another. The consortia catabolized glucose and exchanged either acetic or lactic acid to create producer-consumer food webs. The organic acids had different inhibitory properties and different opportunity costs associated with their positions in central metabolism. The exchanged metabolites modulated different consortial dynamics. The acetic acid-exchanging (AAE) consortium had a "push" interaction motif where acetic acid was secreted faster by the producer than the consumer imported it, while the lactic acid-exchanging (LAE) consortium had a "pull" interaction motif where the consumer imported lactic acid at a comparable rate to its production. The LAE consortium outperformed wild-type (WT) batch cultures under the environmental context of weakly buffered conditions, achieving a 55% increase in biomass titer, a 51% increase in biomass per proton yield, an 86% increase in substrate conversion, and the complete elimination of by-product accumulation all relative to the WT. However, the LAE consortium had the trade-off of a 42% lower specific growth rate. The AAE consortium did not outperform the WT in any considered performance metric. Performance advantages of the LAE consortium were sensitive to environment; increasing the medium buffering capacity negated the performance advantages compared to WT.

**IMPORTANCE** Most naturally occurring microorganisms persist in consortia where metabolic interactions are common and often essential to ecosystem function. This study uses synthetic ecology to test how different cellular interaction motifs influence performance properties of consortia. Environmental context ultimately controlled the division of labor performance as shifts from weakly buffered to highly buffered conditions negated the benefits of the strategy. Understanding the limits of division of labor advances our understanding of natural community functioning, which is central to nutrient cycling and provides design rules for assembling consortia used in applied bioprocessing.

**KEYWORDS** consortia, division of labor, metabolite inhibition, synthetic ecology

**Ad Hoc Peer Reviewer** Davide Ciccarese, MIT

Address correspondence to Ross P. Carlson, rossc@montana.edu.

*Present address: Kathryn Pintar, Seeq! Inc., St. Paul, Minnesota, USA.

§Present address: Diana Schepens, Whitworth College, Spokane, Washington, USA.

◇Present address: Ashley Schrammeck, Glaxo Smith Kline Inc., Hamilton, Montana, USA.

∞Present address: Timothy Johnson, Bechtel National Inc, Richland, Washington, USA.

‡Present address: Alissa Bleem, Renewable Resources and Enabling Sciences Center, National Renewable Energy Laboratory, Golden, Colorado, USA.

The authors declare no conflict of interest.

Division of labor can enhance the fitness of interacting microorganisms via mechanisms that modulate growth rates or improve biomass yields (1–4). However, constraints on division of labor due to environmental context are largely undocumented. This knowledge

gap includes division of labor strategies such as the exchange of organic acids. Secretion of organic acids in the presence of $O_2$, a phenotype often termed "overflow metabolism," is a common microbial strategy for acclimating to stresses such as imbalances in electron donors and acceptors, imbalances in anabolic and catabolic nutrients, constrained cellular volume, and/or limited cellular surface area (5–11). Therefore, the phenotype a cell uses to acclimate to one environmental stress, like nitrogen limitation, can create additional stresses including high concentrations of inhibitory organic acids. These secreted organic acids can create and influence food webs where microorganisms coexist in interdependent communities (12–18). Ultimately, environment and metabolism are interrelated and can influence each other creating complicated ecological networks (19, 20).

Cross-feeding organic acids has costs for the producer such as the loss of potential cellular energy based on the chemical properties of the exchanged metabolite. The accumulation of organic acids may inhibit the growth of both the producer and consumer due to cytosol acidification, membrane solubilization, or the reduction of thermodynamic driving forces necessary for chemical reactions (21–23). Organic acid exchange can also modulate consortia functioning via the relative rates of metabolite excretion and consumption. A consortium can display a "push" metabolite interaction where the organic acid is secreted faster by the producer than imported by the consumer, or the consortium can demonstrate a "pull" metabolite interaction where the consumer strain imports the metabolite at a comparable rate to its production (24, 25). Environmental context influences the costs of organic acid exchange, yet these costs are poorly characterized especially for environments that are weakly buffered or acidic (26–28). These conditions are relevant to many natural habitats. For example, the pH of the human colon shifts dynamically from pH 5 to 8 as a function of axial position, reflecting the limited buffering capacity of colon contents and a dependency on local microbial metabolism (29). Additionally, aquatic ecosystems including estuarine and freshwater systems have weak pH buffering capacities (30–32), although complex environments such as blood and some humic soils can have higher buffering capacities (33–35).

Natural microbial communities are often complex, comprised of hundreds or thousands of interacting species (36, 37), whereas synthetic consortia can be engineered to have a tractable number of defined phenotypes and interactions (23, 38, 39). Synthetic consortia can illuminate basic ecological properties and mechanisms of interaction that can be extrapolated to natural communities. The ability to control variables within synthetic communities also provides a powerful tool to investigate ecological theories (40). To this end, synthetic communities have been applied to problems of understanding the benefits of cooperation, the role of division of labor in the simultaneous utilization of different sugars, mechanisms for enhanced conversion of cellulose to biofuels, and the role of quorum sensing in coordinating consortial behavior, among others (1, 2, 21, 41, 42).

The present study uses synthetic ecology to test two consortia interaction hypotheses. First, it is hypothesized that environmental context, namely, medium pH buffering capacity, influences whether division of labor, involving organic acid exchange, is a competitive consortial interaction strategy. Secondly, it is hypothesized that not all metabolite exchanges are created equal, as push versus pull metabolite exchanges when combined with environmental constraints will result in different consortial performance. These hypotheses were evaluated by constructing two organic acid exchanging consortia: one consortium was based on acetic acid exchange (AAE) and the other on lactic acid exchange (LAE). Additionally, the hypotheses were tested using a dynamic computational model of cellular interactions that accounted for the rates of organic acid exchange and the inhibitory properties of the exchanged organic acids. Understanding the limits of division of labor strategies as a function of environment advances our understanding of consortia design principles essential for rational control of their catalytic potential.

## RESULTS

**Organic acids and culture pH are major mediators of growth inhibition.** Protonated organic acids can diffuse across cellular membranes and inhibit cell growth (22, 43). Acetic and lactic acids have different $pK_a$ values (4.76 and 3.86, respectively) and different molecular weights (60 and 90 g $mol^{-1}$, respectively); both parameters influence the inhibitory properties

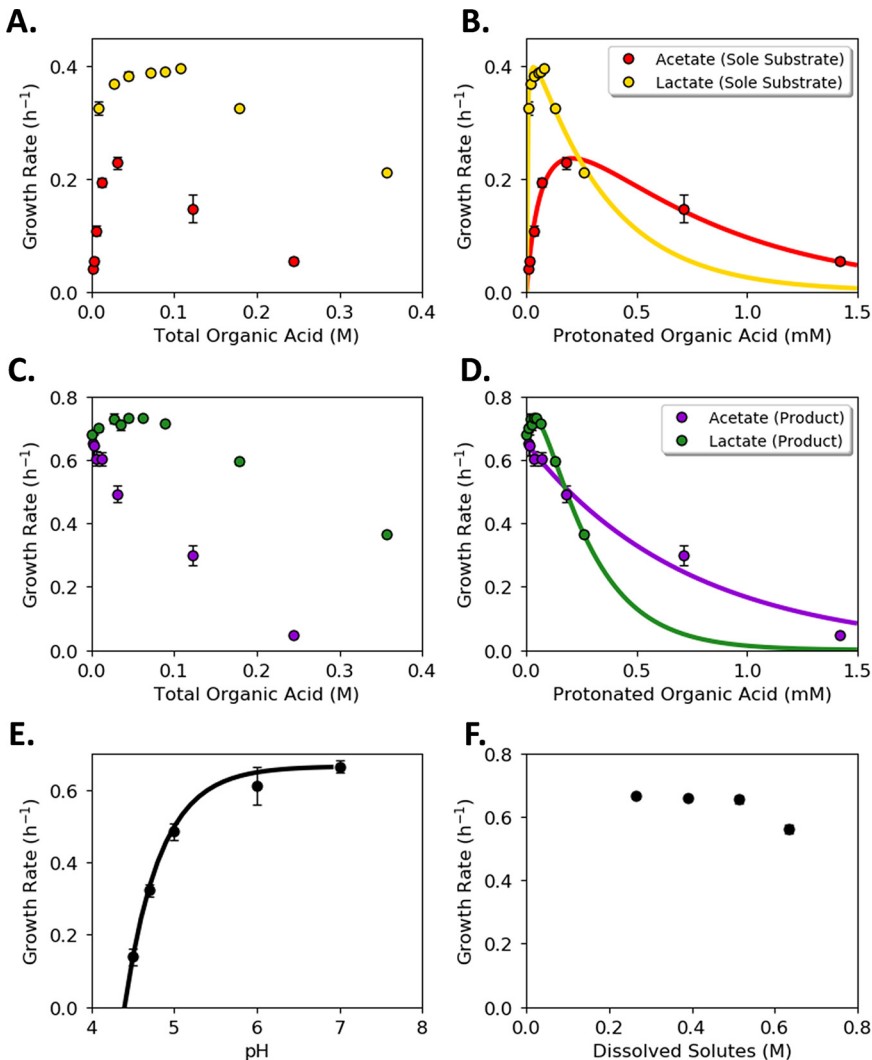

**FIG 1** Inhibition of wild-type *Escherichia coli* cultures as a function of organic acids, pH, and osmotic pressure. (A and B) *E. coli* specific growth rate in conventional M9 medium as a function of acetic acid and lactic acid concentration when the organic acids were the sole substrates. (A) Abscissa plots the total concentration of the metabolite (protonated [HA] and unprotonated [A⁻]). (B) Abscissa plots only the protonated metabolite (HA) concentration. Colored lines represent the specific growth rate predicted with selected models. $R^2 = 0.98$ for acetic acid; $R^2 = 0.93$ for lactic acid. Selected equations and parameters are listed in Table 1. (C and D) *E. coli* specific growth rate in conventional M9 medium as a function of acetic acid and lactic acid concentration when 56 mM glucose was added in addition to the organic acid. (C) Abscissa plots the total concentration of the metabolite (protonated [HA] and unprotonated [A⁻]). (D) Abscissa plots only the protonated metabolite (HA) concentration. Colored lines represent the specific growth rate predicted with selected models. $R^2 = 0.97$ for acetic acid and glucose; $R^2 = 0.98$ for lactic acid and glucose. Selected equations and parameters are listed in Table 1. (E) Specific growth rate of *E. coli* cultures as a function of medium pH. Cultures grown in conventional M9 medium with 56 mM glucose as the sole carbon source (no added organic acids). Line represents the specific growth rate predicted with selected model [$r(3) = 0.998$, $P < 0.001$]. Selected fit equation and parameters are listed in Table 1. (F) Effect of osmotic pressure on specific growth rate of *E. coli* cultures. Growth was tested at 1, 1.5, 2, and 2.5× the concentration of conventional M9 medium components (converted to concentration of dissolved solutes with Visual MINTEQ 3.1). Each point represents the average of at least three biological replicates with error bars representing standard deviation (SD).

of the organic acids (22). The inhibitory effects of acetic and lactic acids on wild-type *Escherichia coli* K12 MG1655 were quantified using two different contexts. First, the inhibitory properties of the organic acids were measured when the metabolites were present as the sole reduced carbon source in M9 medium, and second, when the organic acids were present along with glucose in M9 medium (Fig. 1). The inhibitory properties were quantified using specific growth rate and plotted against (i) the total organic acid concentration ([HA + A⁻], where HA is the protonated organic acid and A⁻ is the base), and (ii) the protonated organic acid

**TABLE 1** Selected expressions and parameters used to model organic acid growth and inhibition kinetics for wild-type *Escherichia coli*[a]

| Condition | Best-fit expression | Parameters |
|---|---|---|
| Acetic acid as sole substrate | $\mu = \mu_m \left( \dfrac{A}{K_A + A} \right) e^{-\frac{A}{K_I}},$ where $\mu$ is growth rate ($h^{-1}$) and $A$ is concn of protonated acetate (mM) | $\mu_m = 0.4\ h^{-1}$ $K_A = 0.0723$ mM $K_I = 0.760$ mM $R^2 = 0.98$ |
| Lactic acid as sole substrate | $\mu = \mu_m \left( \dfrac{L}{K_L + L} \right) e^{-\frac{L}{K_I}},$ where $\mu$ is growth rate ($h^{-1}$) and $L$ is concn of protonated lactate (mM) | $\mu_m = 0.5\ h^{-1}$ $K_L = 0.0038$ mM $K_I = 0.317$ mM $R^2 = 0.93$ |
| Acetic acid with glucose | $\mu = \mu_m \left( \dfrac{G}{K_G + G} \right) e^{-KA},$ where $\mu$ is growth rate ($h^{-1}$), $G$ is concn of glucose, and $A$ is concn of protonated acetate (mM) | $\mu_m = 0.65\ h^{-1}$ $K_G = 0.005$ mM $K = 1.35\ mM^{-1}$ $R^2 = 0.97$ |
| Lactic acid with glucose | $\mu = \mu_m \left[ \left( \dfrac{G}{K_G + G} \right) + \left( \dfrac{L}{K_L + L} \right) \right] e^{-\alpha L},$ where $\mu$ is growth rate ($h^{-1}$), $G$ is concn of glucose (set to 56 mM), and $L$ is concn of protonated lactate (mM) | $\mu_m = 0.65\ h^{-1}$ $K_G = 0.005$ mM $K_L = 0.0743$ m, $\alpha = 4.44\ mmol^{-1}$ $R^2 = 0.98$ |
| pH | $\mu = \mu_m \left( 1 - \dfrac{H}{H^*} \right),$ where $\mu$ is growth rate ($h^{-1}$), and $H$ is concn of protons (M) | $\mu_m = 0.665\ h^{-1}$ $H^* = 10^{-4.4}$ M (critical threshold above which growth is not possible) |

[a]See Fig. S1 for additional equations and github for Python code and data.

concentration only ([HA]; calculated using the Henderson-Hasselbalch equation). Both acetic and lactic acid supported *E. coli* growth in M9 medium as the sole substrate; specific growth rates increased with increasing organic acid concentrations up to a critical threshold, after which further increases in organic acid reduced the specific growth rate. The maximum specific growth rate on acetic acid was ~0.23 $h^{-1}$ at a total concentration of ~20 mM ([HA + A$^-$]), while the maximum specific growth rate on lactic acid was ~0.4 $h^{-1}$ at a total concentration of ~100 mM (Fig. 1A). Acetic acid was inhibitory to growth at lower total concentrations than lactic acid (Fig. 1A). However, the protonated, uncharged form of the organic acids is a major mediator of inhibition due to enhanced diffusion across cellular membranes (22). When the specific growth rates were plotted as a function of the protonated organic acid concentrations ([HA]), lactic acid was more inhibitory than acetic acid at higher concentrations (Fig. 1B).

Adding glucose (56 mM) to the M9 medium changed the metabolic role of acetic acid (Fig. 1C). *E. coli* preferentially consumed glucose as the carbon and energy source while acetic acid served as an inhibitor. The specific growth rates of the cultures in the presence of glucose and acetic acid never exceeded the specific growth rate of the glucose-only medium. However, mixtures of glucose and lactic acid increased the specific growth rates of the cultures over glucose-only medium (Fig. 1C). The maximum specific growth rate (~0.73 $h^{-1}$) peaked at a total lactic acid concentration of ~80 mM, after which the specific growth rate decreased. Acetic acid was more inhibitory than lactic acid when examined on a total concentration basis in the presence of glucose (Fig. 1C). However, examination of the data on a protonated acid basis quantified the more inhibitory nature of lactic acid at higher concentrations (Fig. 1D).

pH and osmotic pressure were also investigated as separate environmental parameters. Growth inhibition, quantified as culture specific growth rate, was measured for each variable. Specific growth rate for *E. coli* cultures grown in M9 medium with glucose decreased as the pH decreased (Fig. 1E). The specific growth rate was minimally affected over the tested range of osmotic pressures, which represented 1 to 2.5× M9 salts with 56 mM glucose (Fig. 1F).

Organic acid and pH inhibition equations were parameterized using culturing data. A variety of kinetic equations for modeling inhibitory compounds was tested (44). Many of the equations fit the data well (Fig. S1). Selected fits for each organic acid and condition are presented in Table 1. Predicted values are plotted against experimental data in Fig. 1B [$r(5) = 0.992$, $P < 0.001$ for acetic acid; $r(6) = 0.967$, $P < 0.001$ for lactic acid] and Fig. 1D

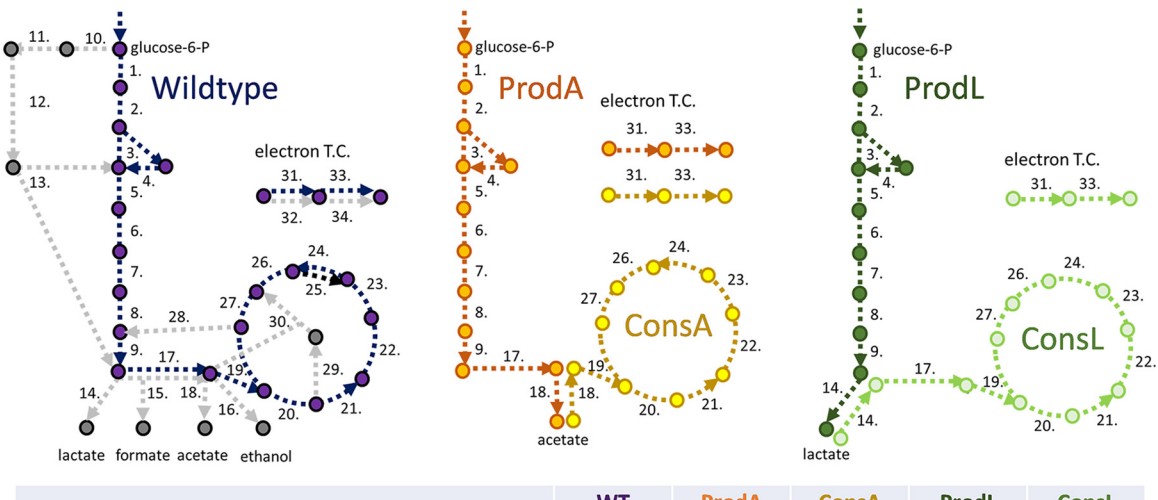

| | WT | ProdA | ConsA | ProdL | ConsL |
|---|---|---|---|---|---|
| **Catabolic efficiency (ATP (mol substrate)$^{-1}$)** | 26 | 4 | 6 | 2 | 12 |
| **Catabolic efficiency (Cmol substrate (ATP)$^{-1}$)** | 0.23 | 1.5 | 0.29 | 3 | 0.25 |
| **Opportunity cost (ATP (mol glucose)$^{-1}$)** | | 22 | | 24 | |
| **Total proteome investment (Amino Acids (AA))** | 101,548 | 63,424 | 49,047 | 13,662 | 89,646 |
| **Total proteome investment (ATP equivalents)** | 2,597,136 | 528,533 | 1,409,397 | 227,700 | 2,390,560 |
| **Normalized proteome investment (AA (ATP)$^{-1}$)** | 3,906 | 15,856 | 8,175 | 6,831 | 7,471 |
| **Normalized proteome investment (ATP equiv. (ATP)$^{-1}$)** | 99,890 | 132,133 | 234,899 | 113,850 | 199,213 |

**FIG 2** Representations of the central metabolism of the five *E. coli* phenotypic guilds: generalist (wild-type [WT]), producer secreting acetic acid (ProdA), consumer catabolizing acetic acid (ConsA), producer secreting lactic acid (ProdL), and consumer catabolizing lactic acid (ConsL). Metabolite exchanges necessitate costs which are translated into fitness considerations for the five guilds including catabolic efficiency of relevant substrate, ATP opportunity costs for a guild phenotype, and central metabolism proteome investment costs. Numbers in figures refer to enzyme catalyzed reactions, see Table S1 and S2 for key. AA, amino acids; Cmol, carbon mole.

[$r(5) = 0.991$, $P < 0.001$ for acetate; $r(7) = 0.998$, $P < 0.001$ for lactate]. Additional results and code can be found in the supplemental information (Fig. S1). Equations reported in the literature fit the culture properties for acetic acid as the sole substrate, for combinations of acetic acid and glucose, and for lactic acid as the sole substrate (44). The combination of lactic acid and glucose required a dual substrate equation with an inhibition term to model the data well (Fig. 1D and Fig. S1).

A pH inhibition equation, separate from organic acids, was also fit to the experimental data [$r(3) = 0.998$, $P < 0.001$; Fig. 1E and Table 1]. The parameterized equations considering organic acids, glucose, and pH were combined into a predictive model for *E. coli* growth under multiple environmental stresses. The model was tested with an independent experimental data set quantifying *E. coli* growth in the presence of glucose with varying concentrations of acetic acid, different initial pH values, and different osmotic pressures. The predicted values matched the experimental data well, with a Pearson's correlation coefficient of 0.99 and corresponding $P$ value $\ll 0.01$ (Fig. S2).

**Producer-consumer consortia built to exchange either acetic or lactic acids.** Acetic acid and lactic acid display different inhibitory properties, support different specific growth rates, confer different amounts of chemical energy, and require different enzymatic pathway investments for their production or consumption (Fig. 1 and 2). Two *E. coli*-derived synthetic

**TABLE 2** Physiological properties of *E. coli* generalist and producer/consumer guilds in monoculture[a]

| | Generalist (WT) | Acetate producer (ProdA) | Lactate producer (ProdL) | Consumer (ConsA/ConsL) |
|---|---|---|---|---|
| Carbon source | Glucose ($n = 4$) | Glucose ($n = 3$) | Glucose ($n = 9$) | Acetic acid ($n = 2$)/lactic acid ($n = 9$) |
| Maximum growth rate | 0.65 h$^{-1}$ ± 0.01 | 0.54 h$^{-1}$ ± 0.007 | 0.24 h$^{-1}$ ± 0.003 | 0.14 h$^{-1}$ ± 0.003/0.41 h$^{-1}$ ± 0.01 |
| Biomass yield | 0.43 g biomass per g glucose ± 0.02 | 0.20 g biomass per g glucose ± 0.01 | 0.05 g biomass per g glucose ± 0.001 | 0.24 g biomass per g acetic acid ± 0.00/ 0.48 g biomass per g lactic acid ± 0.01 |
| Organic byproduct yield | 0.11 g acetic acid per g glucose ± 0.01 | 0.34 g acetic acid per g glucose ± 0.004 | 0.86 g lactic acid per g glucose ± 0.01 | NA |

[a]The consumer guild is capable of growth on glucose but at a very low rate (0.016 h$^{-1}$). The lactic acid producer accumulates minor amounts of acetic acid (<0.06 g/L) and succinic acid (<0.12 g/L) in addition to lactic acid during stationary phase, which aligns with results reported in reference 48. WT, wild-type; ProdA, producer secreting acetic acid; ProdL, producer secreting lactic acid; ConsA, consumer catabolizing acetic acid.

consortia were designed to test different organic acid exchange strategies hypothesized to change with both environmental context and interaction motif (push or pull metabolite exchange). The consortia were assembled by combining pairs of *E. coli* strains engineered for different phenotypes termed guilds: the producer guilds catabolized glucose and produced either acetic acid or lactic acid, and a consumer guild catabolized the organic acids (45, 46).

The producer guilds were designed for overflow phenotypes analogous to published metabolisms for either nitrogen- or iron-limited *E. coli* growth (47). The producer strain specializing in acetic acid secretion (ProdA) was created using four gene deletions (Δ*aceA* Δ*ldhA* Δ*frdA* Δ*atpF*). The Δ*atpF* gene deletion resulted in elevated substrate-level phosphorylation and high acetic acid secretion from glucose catabolism, analogous to other published systems (48, 49). A second producer guild specializing in lactic acid secretion (ProdL) had four respiration-associated operons deleted (Δ*cbdAB* Δ*cydAB* Δ*cyoABCD* Δ*ygiN*) (50). The consumer strain catabolized either acetic or lactic acid (ConsA/ConsL), but not glucose, and was constructed by deleting four genes associated with glucose transport and phosphorylation (Δ*ptsG* Δ*ptsM* Δ*glk* Δ*gcd*) (46). Wild-type *E. coli* K12 MG1655 (WT) was used as the generalist guild for all comparisons. Monocultures of each guild were characterized in conventional M9 medium with their respective substrates to quantify their physiology (Table 2).

**Catabolic efficiency and opportunity costs differ among the guilds.** Cellular energy production was analyzed for all five guilds: (i) generalist (WT), (ii) acetic acid producer (ProdA), (iii) lactic acid producer (ProdL), (iv) consumer based on acetic acid oxidation (ConsA), and (v) consumer based on lactic acid oxidation (ConsL) using a published *E. coli* metabolic model (51, 52) (Fig. 2 and Table S1 and S2). ProdA produced 4 mol ATP (mol glucose)$^{-1}$ while ProdL produced 2 mol ATP (mol glucose)$^{-1}$. ProdA extracted more energy from glucose via substrate-level phosphorylation using the Pta enzyme. The exchanged organic acids have the same degree of reduction (4 electron mol Cmol$^{-1}$) but different chemical energies (quantified here as enthalpy of combustion ($\Delta H_c^\circ$) = 875 and 1,362 kJ mol$^{-1}$ for acetic acid and lactic acid, respectively), largely due to the difference in the molecular weights of the molecules. Additionally, the molecules entered central metabolism at different positions, influencing their potential for substrate level phosphorylation. ConsA and ConsL produced 7 and 12 mol ATP (mol organic acid)$^{-1}$, respectively. The *in silico* model of the WT generalist produced 26 mol ATP (mol glucose)$^{-1}$ when the substrate was completely oxidized (Fig. 2 and Table S2).

Cross-feeding by-products, such as organic acids, necessarily entails opportunity costs for the producer guild. Here, opportunity costs are a quantification of benefits not realized by a cell due to use of a particular phenotype (5, 24). The exchange of a reduced metabolite precluded its use by the producer guild for other functions such as cellular energy generation. Opportunity costs were quantified based on cellular energy that was not generated due to metabolite secretion (51, 52) (Fig. 2 and Table S2). The opportunity cost for ProdA was 22 mol ATP (mol glucose)$^{-1}$ while the opportunity cost for ProdL was 24 mol ATP (mol glucose)$^{-1}$.

**Proteomic investment costs for each guild were heavily influenced by a few enzyme-catalyzed reactions.** Metabolic phenotypes require the assembly of the necessary proteomes (51, 52). An *in silico* analysis quantified the amino acid requirements to realize the core proteomes of the five guilds. *In silico* analysis considered the minimum proteome proxy for modeling the relationship between flux and enzyme concentration, as described previously (51, 52). This proxy assumes that the concentration of all central metabolism enzymes

can be approximated by the relationship $[E_i]/[E_j] \sim 1$; this proxy has been applied with notable successes in *E. coli* as well as other microbial species (52, 53).

The generalist, with its complete oxidation metabolism, required the largest total proteome investment of $\sim$97,000 amino acids although it had the smallest proteome investment per ATP produced at 3,906 amino acids ATP$^{-1}$ (Fig. 2). In descending order based on total investment, ConsL had an investment cost of $\sim$85,000 amino acids (7,471 amino acids ATP$^{-1}$), ProdA required $\sim$59,000 amino acids (15,856 amino acids ATP$^{-1}$), ConsA required $\sim$44,000 amino acids (8,175 amino acids ATP$^{-1}$), and finally, ProdL required the smallest investment of $\sim$14,000 amino acids (6,831 amino acids ATP$^{-1}$). The pyruvate dehydrogenase complex (42,096 amino acids per complex), which oxidizes pyruvate to acetyl-CoA, had a large influence on the investment cost for WT, ConsL, and ProdA (51, 52). The investment cost from this enzyme was avoided by the ConsA and ProdL guilds. There was also a large investment cost associated with the citric acid cycle due largely to the $\alpha$-ketoglutarate dehydrogenase complex and the electron transport chain enzymes (Table S1 and S2).

The different guilds have different core proteomes that catabolize different substrates with different metabolic efficiencies complicating inter-guild comparisons. Therefore, the proteome investment required for each guild was converted into ATP equivalents using the guild-specific *in silico* models (54) (Table S4). The carbon mols (Cmols) of substrate required for each guild to assemble the core proteome were calculated; this quantity of substrate was then converted into an equivalent number of ATP using the guild-specific Cmol substrate (mol ATP)$^{-1}$ yield. The proteome resource requirements were calculated as a total investment of ATP equivalents to construct the proteome, and additionally, this investment was normalized to the number of ATP produced per core metabolism (Fig. 2). The WT proteome had the most efficient ratio of proteome investment per ATP produced [99,890 ATP equivalents invested (ATP produced)$^{-1}$], followed by ProdL, ProdA, ConsL, and finally ConsA which required a proteome investment equivalent of 234,899 ATP (ATP produced)$^{-1}$.

**Experimental properties of organic acid exchanging consortia.** Batch growth properties of the producer guilds were measured as monocultures and as cocultures with the consumer guild. Glucose was the sole reduced carbon source in the modified M9 medium, and four initial pH values were tested (6.0, 6.5, 7.0, and 7.5). The modified M9 medium contained 6.3 mM total phosphate to represent an environment with a low pH buffering capacity. WT monocultures served as generalist controls.

WT monocultures produced the highest final biomass titer (mean = 0.31 $\pm$ 0.01 g cell dry weight [cdw] L$^{-1}$; $n$ = 3) and displayed the highest specific growth rate at an initial pH of 7.0 (mean = 0.53 $\pm$ 0.01 h$^{-1}$; $n$ = 3; Fig. 3A and B). The cultures grown with an initial pH of 7.5 had the slowest specific growth rate (mean = 0.37 $\pm$ 0.01 h$^{-1}$; $n$ = 3; $P \ll 0.05$ compared with pH 7.0 results; Fig. 3B). Medium pH decreased with biomass accumulation for all four initial pH values (Fig. 3C). The WT cultures accumulated acetic acid consistent with an *E. coli* overflow metabolism (Fig. 3D). Growth data and calculated rates for all conditions and guilds are provided in Table S3.

**Acetic acid-exchanging consortium displayed a push interaction motif.** Biomass titers for the acetic acid-exchanging (AAE) consortium changed with initial medium pH (Fig. 4A). The highest biomass titer occurred when the initial pH of the medium was 7.5 (mean = 0.12 $\pm$ 0.00 g cdw L$^{-1}$; $n$ = 3), whereas the highest specific growth rate occurred when the initial pH of the medium was 7.0 (mean = 0.27 $\pm$ 0.01 h$^{-1}$; $n$ = 3; Fig. 4A and B). The consortium growth rate was an aggregate rate comprised of the ProdA and ConsA growth rates and was slower than the ProdA monoculture (Fig. 4B and Table 2). The culture pH decreased with time in trends proportional to biomass production with a final pH range of 4.1 to 4.5 (Fig. 4C). The AAE consortium accumulated 50% more biomass relative to ProdA monocultures at initial pH values of 6.0 (mean = 0.08 $\pm$ 0.01 g cdw L$^{-1}$ and mean = 0.05 $\pm$ 0.00 g cdw L$^{-1}$, respectively; $n$ = 3; $P < 0.05$) and 7.5 (mean = 0.12 $\pm$ 0.00 g cdw L$^{-1}$ and mean = 0.09 $\pm$ 0.01 g cdw L$^{-1}$, respectively; $n$ = 3; $P < 0.05$), quantifying a benefit of guild interactions within the tested environs (Fig. 4D).

The rate of acetic acid secretion by ProdA exceeded the rate of consumption by ConsA (Fig. 4E), resulting in organic acid accumulation and a cellular interaction motif termed here as a metabolite "push." The presence of ConsA did not result in the complete consumption

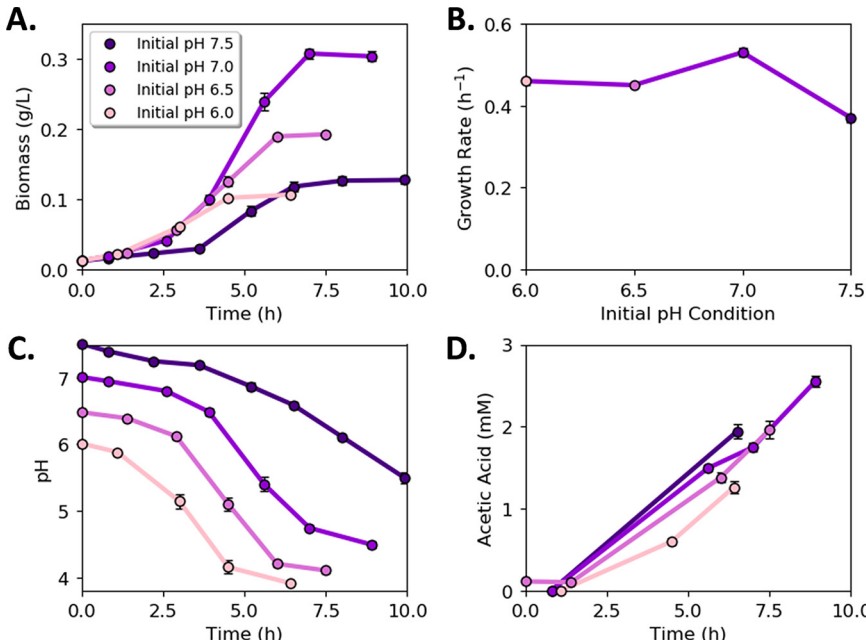

**FIG 3** Wild-type culture (generalist guild) properties as a function of initial medium pH. (A) Biomass accumulation with time as a function of initial medium pH. (B) Specific growth rate as a function of initial medium pH. (C) pH as a function of time for cultures starting at different initial pH values. (D) Acetic acid concentration as a function of time for cultures starting at different initial pH values. Fewer data points are presented for acetic acid as metabolite samples were not measured at every sampling point to conserve culture volume. Error bars represent the standard deviation (SD) of three biological replicates.

of secreted acetic acid due to low pH and high concentrations of acetic acid arresting guild growth. The cellular ratio of the two guild members was measured as a function of initial pH and cultivation time (Fig. 4F). The cellular ratio was approximately 1 ProdA:1 ConsA at the cessation of growth regardless of the initial pH (mean = $1.10 \pm 0.27$; $n = 12$).

**Lactic acid-exchanging consortium displayed a pull interaction motif.** Batch growth of the lactic acid-exchanging (LAE) consortium produced the highest biomass titers when the initial pH of the medium was 7.0 (mean = $0.48 \pm 0.02$ g cdw $L^{-1}$; $n = 3$; Fig. 5A). The highest specific growth rates occurred when the initial pH of the medium was 6.5 and 7.0 (mean = $0.21 \pm 0.01$ $h^{-1}$ and mean = $0.22 \pm 0.01$ $h^{-1}$, respectively; $n = 3$; $P > 0.05$; Fig. 5B). Culture pH decreased continuously with time to a final endpoint of ~4, except for the LAE consortium which had an initial pH of 7.5 (Fig. 5C). The pH of this culture initially decreased to ~5 before recovering to ~6.8. This property was analyzed in more detail in later sections. The LAE consortium had substantially higher biomass titers and biomass per glucose yields compared to ProdL monocultures (Fig. 5D). In contrast to the AAE consortium, the LAE consortium grew faster than the ProdL monoculture, likely due to the high ConsL specific growth rate (Table 2).

The LAE consortium had balanced rates of lactic acid secretion and consumption, resulting in low accumulation of the organic acid (Fig. 5E). The consortial interaction template was termed a metabolite "pull" mechanism because ConsL imported the organic acid at rates comparable to the ProdL secretion rates. The secreted lactic acid was nearly exhausted by the end of the growth phase; any remaining lactic acid was consumed during stationary phase. Selective agar plating quantified the cellular ratios of the ProdL and ConsL guilds (Fig. 5F). The producer ratio decreased during growth, reaching an average ratio of 5 ProdL:95 ConsL by stationary phase (mean = $0.04 \pm 0.11$; $n = 9$).

**LAE consortium properties vary as a function of initial cell ratios.** The properties of the LAE consortium were studied for pH robustness and optimality of biomass titer. Different initial cellular ratios of ProdL:ConsL were tested (1:1, 10:1, 100:1, and 1:2) using an initial medium pH of 7.0 and 7.5. For these experiments, ProdL was inoculated at the same concentration as the preceding 1:1 cell ratio experiments (Fig. 5), while the ConsL concentration

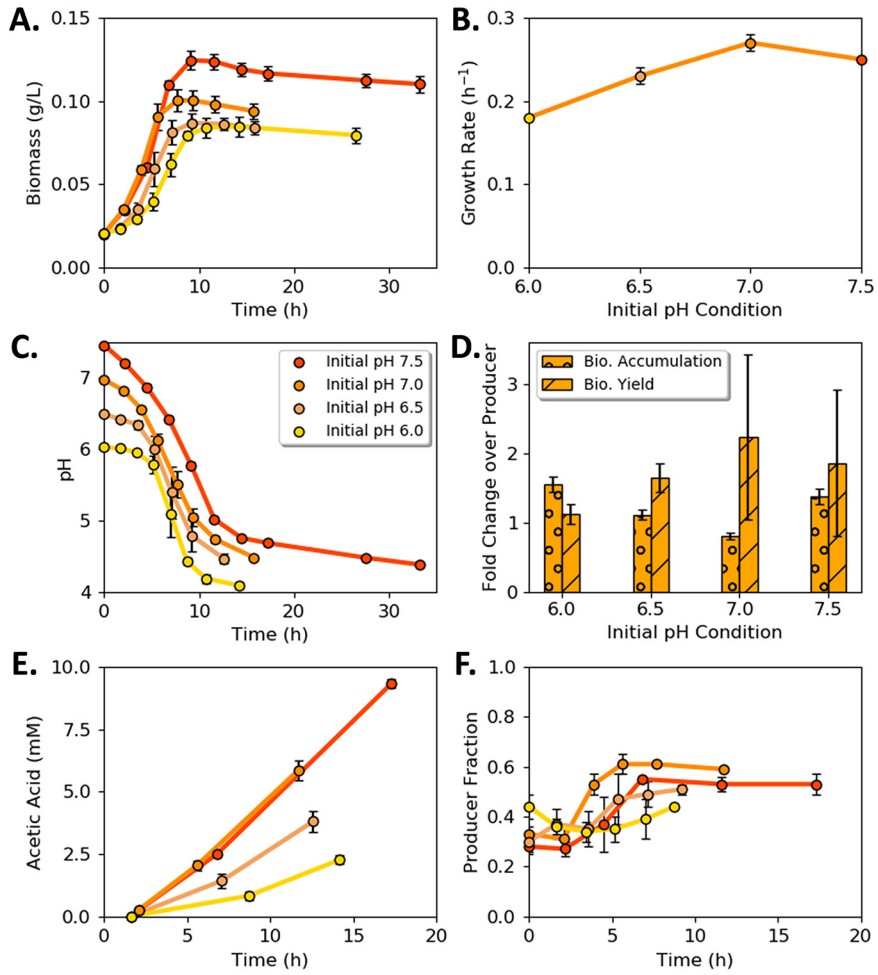

**FIG 4** Acetic acid exchanging (AAE) consortium as a function of initial pH. (A) Consortium biomass concentration with time for four different initial pH values. (B) Consortium specific growth rate as a function of initial medium pH. (C) Medium pH with time as a function of initial medium pH. (D) Comparison of AAE consortium to acetic acid producer (ProdA) monoculture as a function of initial pH. (E) Acetic acid concentration with time as a function of initial medium pH. (F) Cell fraction of ProdA guild to ConsA guild with time as a function of initial medium pH. Fewer data points are presented for acetic acid and cell fraction as samples were not measured at every sampling point to conserve culture volume. Data are mean of three biological replicates with error bars representing standard deviation (SD).

was adjusted to achieve the desired cell ratios. This design resulted in a near constant volumetric lactic acid production rate across the conditions, with the primary experimental change being the ConsL cell concentration.

The consortium behavior was sensitive to the initial conditions. Biomass titers were highest for an initial 1:1 guild ratio and an initial pH of 7.0 (mean = 0.48 ± 0.02 g cdw L$^{-1}$; $n = 3$; Fig. 6A); however, this condition did not exhibit a pH recovery trend. Different initial guild ratios were required to induce the pH recovery trend when the initial pH was 7.0 (Fig. 6B). Cultures with either higher or lower ProdL:ConsL cell ratios (100:1, 10:1, and 1:2) modulated the pH trajectory after an early pH decrease, demonstrating a pH recovery phase corresponding to lactic acid consumption (Fig. 6B). Minimal, transient accumulation of lactic acid was observed before its subsequent consumption (Fig. 6C). Small amounts of acetic acid accumulated during the pH 7.5 experiments (Fig. 6E and Table S3). The consumer was postulated to convert the lactic acid into acetic acid, which was then later consumed in a diauxic manner. ConsL dominated the consortium by the end of batch growth for all tested conditions (Fig. 6D).

Due to the slow growth rate during the pH recovery phase, the LAE consortium experiments lasted up to two weeks. Abiotic controls quantified the role of evaporation on

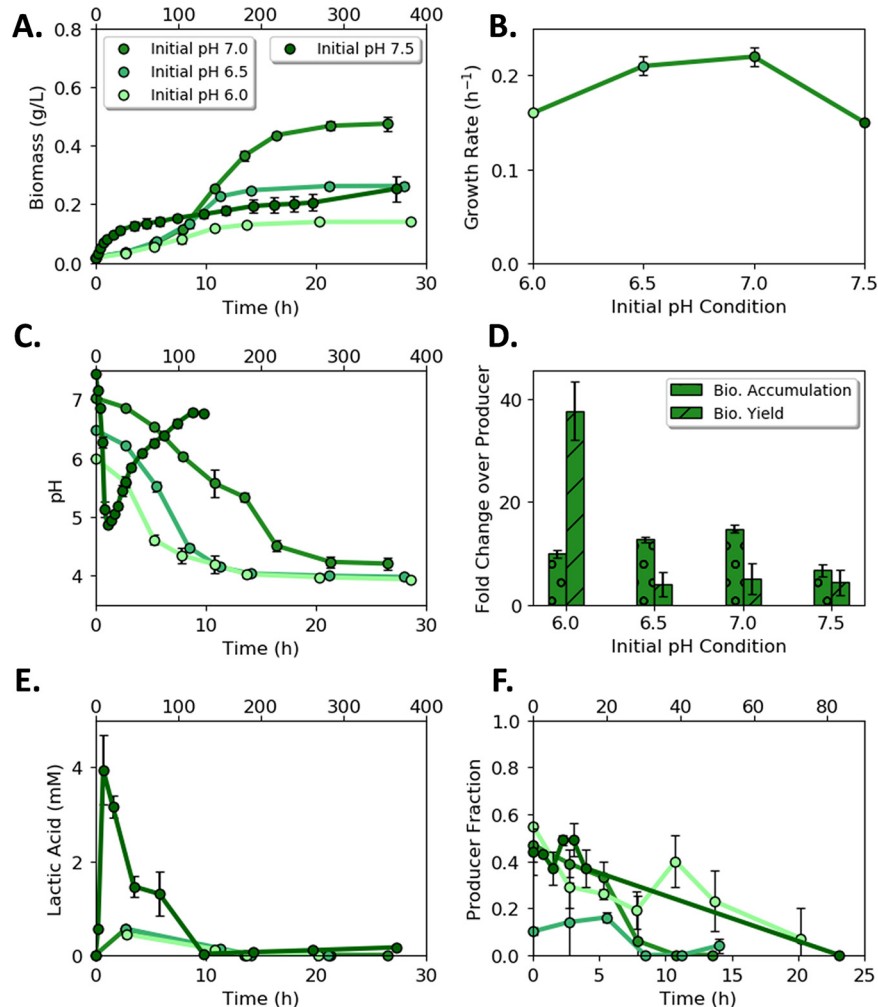

**FIG 5** Lactic acid exchanging (LAE) consortium as a function of initial pH. (A) Consortium biomass concentration with time for four different initial pH values. Note the upper time axis for the culture with an initial pH of 7.5. (B) Consortium specific growth rate as a function of initial medium pH. (C) pH with time as a function of initial medium pH. (D) Comparison of LAE consortium to lactic acid producer (ProdL) monoculture as a function of initial pH. (E) Lactic acid concentration with time as a function of initial medium pH. (F) Cell fraction of ProdL guild to ConsL guild with time as a function of initial medium pH. Fewer data points are represented for lactic acid and cell fraction as samples were not measured at every sampling point to conserve culture volume. Data are mean of three biological replicates with error bars representing standard deviation (SD).

medium volume and changes in pH (Table S4). Evaporation accounted for less than a 10% change in liquid volume and did not result in a change to the medium pH.

**Consortium performance was a function of environmental context.** The synthetic consortia properties were compared to the WT generalist to quantify advantages and disadvantages of the different metabolite exchange templates (Fig. 7). Six metrics of performance were quantified: (i) final biomass titer, (ii) total glucose catabolized, (iii) biomass produced per glucose consumed, (iv) biomass produced per $H^+$ accumulated, (v) accumulation of by-products, and (vi) specific growth rate.

The AAE consortium did not outperform the WT generalist in any considered performance metric, although the biomass per glucose yields of the AAE consortium and generalist were equivalent when the initial medium pH was 7.0 or 7.5.

The LAE consortium outperformed the WT generalist at four of the six performance metrics for a starting pH of 7.0. The LAE consortium produced 55% more total biomass (mean = $0.48 \pm 0.02$ g cdw L$^{-1}$ and mean = $0.31 \pm 0.01$ g cdw L$^{-1}$, respectively; n = 3; P << 0.05), consumed 86% more glucose (mean = $5.38 \pm 0.8$ mmol and mean = $2.89 \pm 0.35$ mmol,

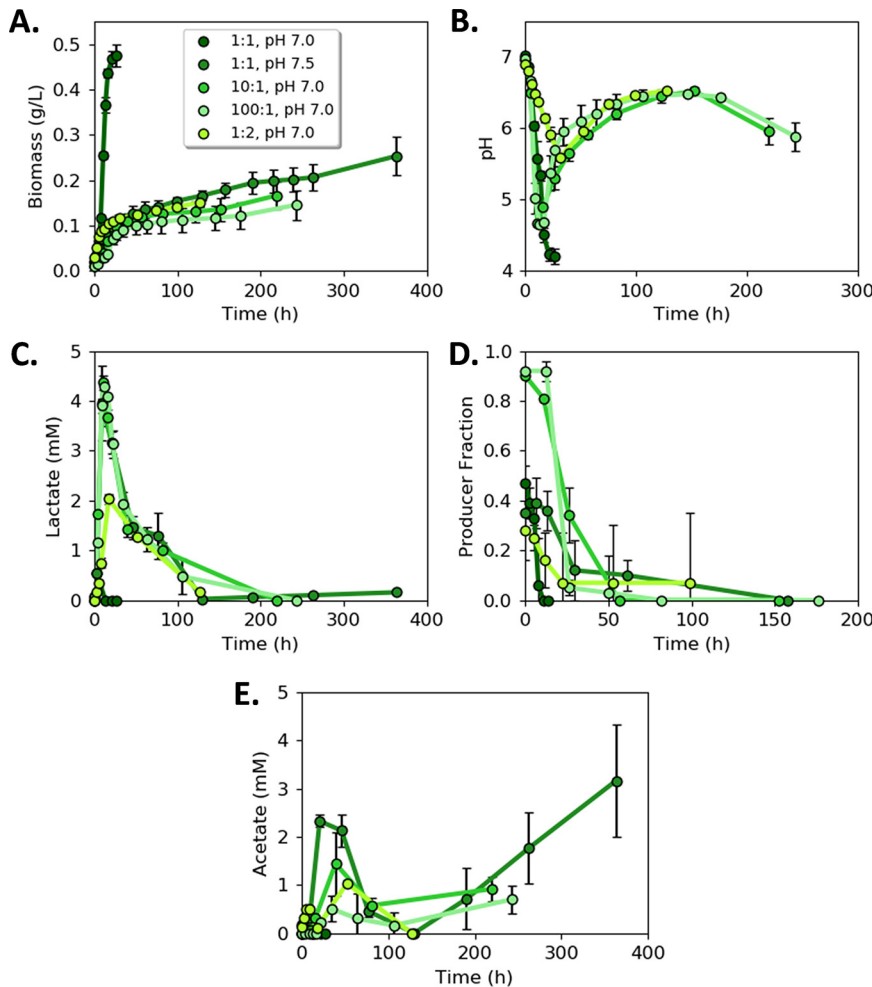

**FIG 6** Lactic acid exchanging (LAE) consortium as a function of different initial guild ratios. (A) Consortium biomass concentration with time for different initial guild cell ratios. (B) Medium pH with time as a function of different initial guild cell ratios. (C) Lactic acid concentration with time for different initial guild cell ratios. (D) Cell fraction of ProdL guild to ConsL guild with time as a function of initial cellular ratios. (E) Acetic acid concentration with time as a function of initial guild ratios. Data are mean of three biological replicates with error bars representing standard deviation (SD).

respectively; $n = 3$; $P < 0.05$), and produced 51% more biomass per $H^+$ secreted compared to the WT generalist [mean = 2.16 $\pm$ 0.02 g cdw (mol $H^+$)$^{-1}$ and 1.43 $\pm$ 0.06 g cdw (mol $H^+$)$^{-1}$, respectively; $n = 3$; $P \ll 0.05$; Fig. 7A to C]. Additionally, the LAE consortium had minimal by-products at stationary phase whereas the WT culture accumulated $\sim$2.5 mM acetic acid. The LAE consortium and WT cultures had comparable biomass per glucose yields (Fig. 7D); however, the WT generalist grew faster (mean = 0.53 $\pm$ 0.01 h$^{-1}$ versus mean = 0.22 $\pm$ 0.01 h$^{-1}$; $n = 3$; $P \ll 0.05$) than the LAE consortium (Fig. 3B and Fig. 5B).

The performance of the LAE consortium was also compared to the WT generalist when grown in M9 medium with a conventional 64 mM phosphate buffer (55) and an initial pH of 7.0. Increasing the pH buffering capacity kept the medium pH $\geq$ 6.2 for the WT cultures and $\geq$ 6.8 for the LAE cultures, which reduced the fitness cost of accumulating organic acids. The pH buffered environment altered the competitive properties of the LAE consortium relative to the WT generalist (Fig. 8). The WT generalist had superior properties in five of the six performance metrics [biomass titer: mean = 1.87 $\pm$ 0.12 g cdw L$^{-1}$ versus 0.55 $\pm$ 0.04 g cdw L$^{-1}$; glucose consumed: mean = 6.49 $\pm$ 0.86 mmol versus 3.92 $\pm$ 0.09 mmol, biomass per glucose yield: mean = 0.42 $\pm$ 0.07 g cdw (g glucose)$^{-1}$ versus 0.18 $\pm$ 0.06 g cdw (g glucose)$^{-1}$; biomass per proton yield: mean = 18.92 $\pm$ 0.52 g cdw (mol $H^+$)$^{-1}$ versus 7.34 $\pm$ 3.49 g cdw (mol $H^+$)$^{-1}$; growth rate: mean = 0.65 $\pm$ 0.00 h$^{-1}$ versus 0.23 $\pm$ 0.02 h$^{-1}$; $n = 3$; $P < 0.05$ for all

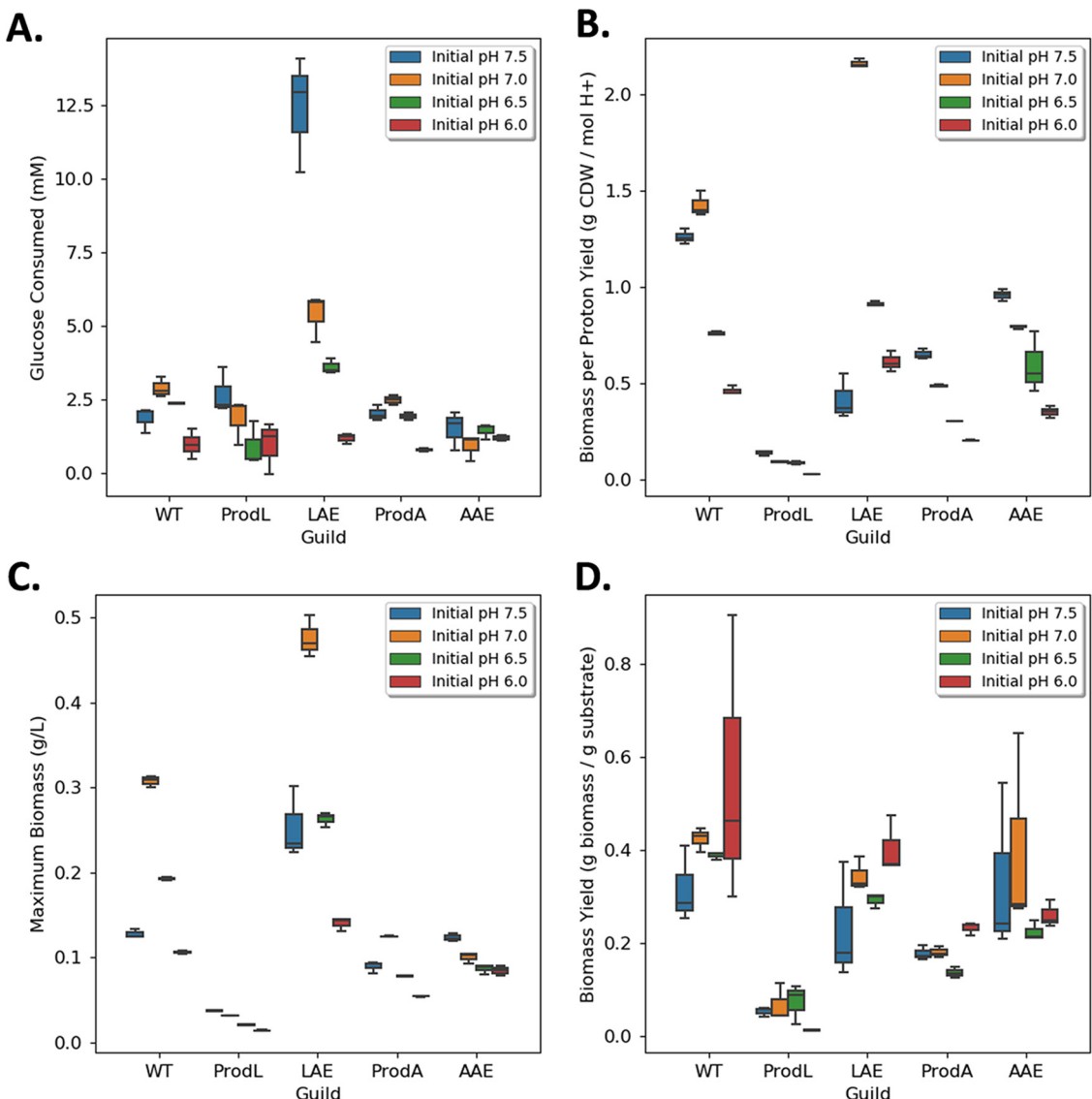

**FIG 7** Summary of performance metrics for organic acid exchanging consortia compared to producer monocultures and wild-type monocultures. (A) Total glucose consumed during batch growth. (B) Biomass (g cdw) produced per mole H$^+$ accumulated in medium. (C) Final biomass concentration (g cdw L$^{-1}$) at stationary phase. (D) Biomass yield on glucose [g cdw (g glucose)$^{-1}$]; cdw, cell dry weight.

comparisons; the LAE consortium had no measurable by-product accumulation at stationary phase], highlighting the environmental context-dependent nature of competitive microbial interactions and optimal consortia design principles.

**Dynamic models predict consortia performance as a function of environment and interaction strategy.** Ordinary differential equation (ODE) models were developed to integrate the experimental data and to test the proposed consortia hypotheses, namely, the role of environmental pH buffering and the role of push versus pull interaction motifs on consortia performance. The ODE models consisted of mass balances on biomass for each guild member, glucose, organic acid, and free protons (pH) and used the growth parameters and the organic acid inhibition equations developed here (Fig. S1 and S3 and Table S3, MATLAB code found at https://github.com/rosspcarlson/becketal-syntheticconsortia). The ODE models were used to generate additional data in support of the proposed division of labor hypotheses. The ODE models were not used to explicitly fit the experimental data because the reported inhibition equations (Table 1 and Fig. S1) were evaluated using wild-type cultures. The synthetic consortia were comprised of strains with numerous gene deletions that could result in

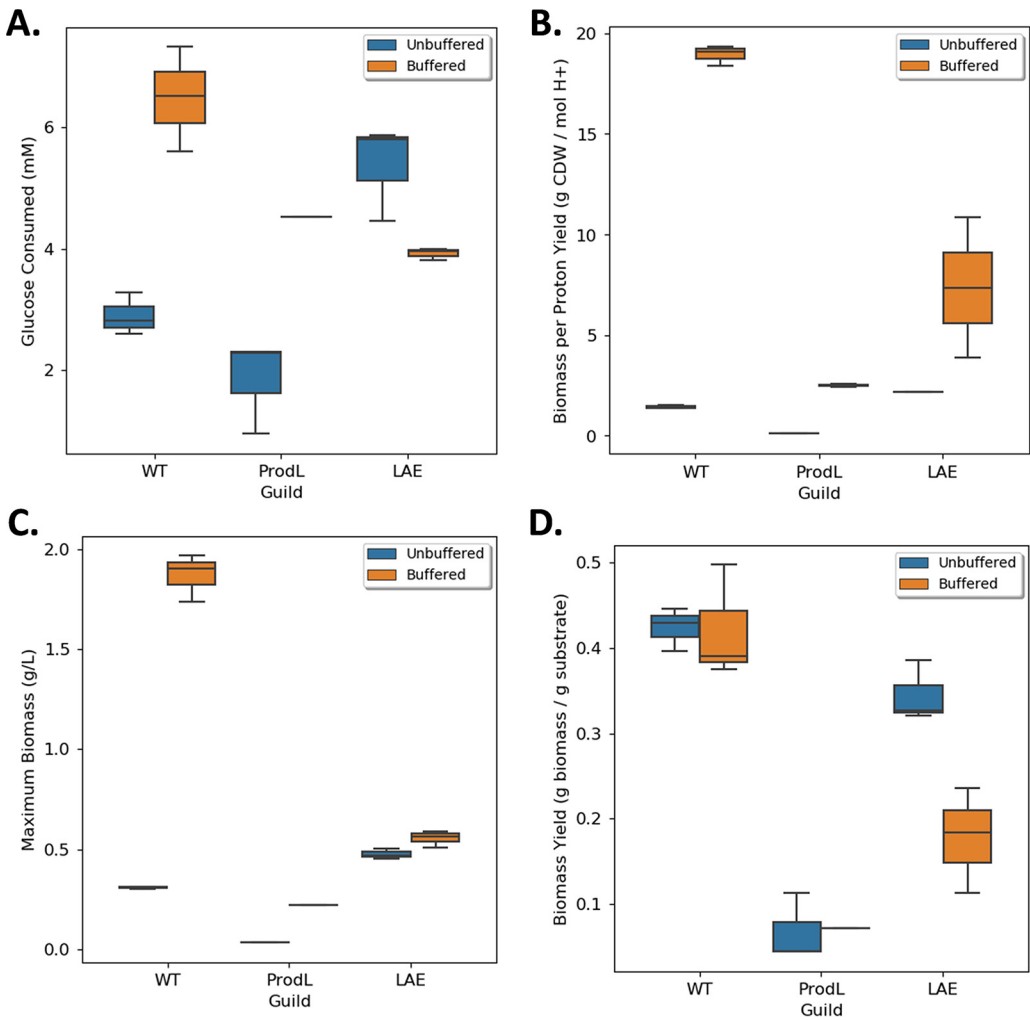

**FIG 8** Performance metrics for lactic acid exchanging (LAE) consortium compared to wild type generalist in unbuffered and highly buffered M9 medium. (A) Total glucose consumed during batch growth. (B) Biomass (g cdw) produced per mole H$^+$ accumulated in medium. (C) Final biomass concentration (g cdw L$^{-1}$) at stationary phase. (D) Biomass yield on glucose [g cdw (pg glucose)$^{-1}$].

pleiotropic effects due to altered metabolisms, redox state, and cellular energy state. For example, the consumer strain was tested for sensitivity to acetic acid and found to be more sensitive than the wild-type likely due to the gene deletions (supplemental data set at https://github.com/rosspcarlson/becketal-syntheticconsortia).

Results from the ODE models supported the hypothesis of environmental context influencing the benefits of division of labor (Fig. 9). At low pH buffering capacity, the organic acid exchanging consortia was predicted to have higher biomass productivity and higher glucose conversion than the wild-type culture (Fig. 9A1 to A4). This is consistent with experimental data (Fig. 7). However, when the pH buffering capacity increased, the model predicted the wild-type culture would grow faster and produce more biomass than the cross-feeding consortia (Fig. 9B1 to B4). Again, the ODE models captured these behaviors and supported the proposed hypothesis (Fig. 8). ODE parameter sets and additional model simulations can be found in Fig. S3.

Results from ODE models also supported the push versus pull cross-feeding hypothesis (Fig. 10 and Fig. S3). Organic acids are inhibitory at elevated concentrations, and the models predicted if the rates of organic acid production and consumption are not balanced, organic acids accumulate to inhibitory levels leading to lower consortium performance (Fig. 10). This is consistent with the push behavior of the AAE consortium and the pull behavior of the LAE consortium (Fig. 4, 5, and 7). Modeling different rates of organic acid production and

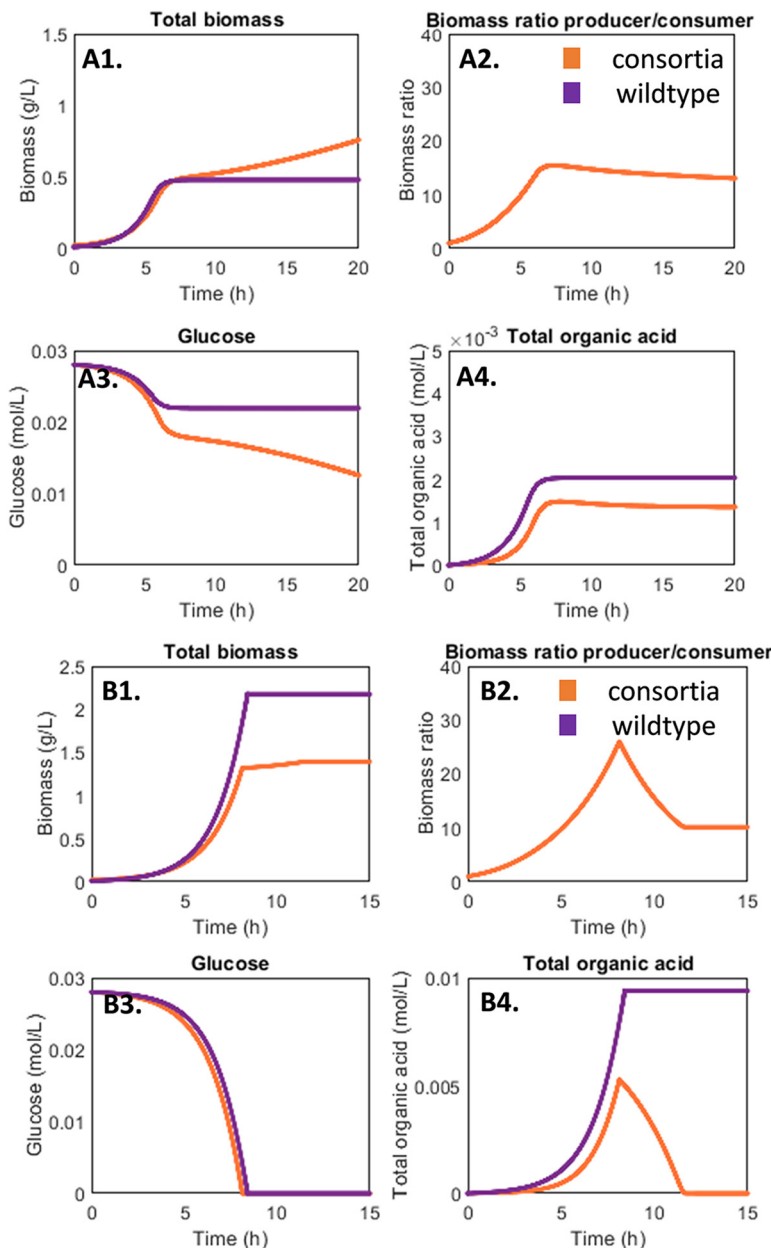

**FIG 9** Dynamic modeling of cross-feeding consortium and wild-type culture performance as a function of environment. (A1 to A4) Quantification of culture performance at low environmental pH buffering capacity (6.3 mM phosphate buffer). (B1 to B4) Quantification of culture performance at higher environmental pH buffering capacity (64 mM phosphate buffer). The cross-feeding consortium outperformed the wild-type culture under the environmental context of low pH buffering capacity, while the wild-type culture grew faster and produced more biomass under the environmental context of higher pH buffering. Models and parameters can be found in Fig. S3 and on github.

consumption can be accomplished by varying growth rates and relevant growth yields. The results presented in the main text were obtained by changing the amount of acetic acid secreted per gram of producer biomass while holding all other parameters, including producer and consumer growth rates, constant. Additional simulations in the supplemental material changed lactic acid secretion rates by changing the producer growth rate (Fig. S3). Simulations demonstrated that both AAE and LAE consortia could result in push or pull interactions with qualitatively similar performance metrics like glucose conversion or biomass titers. However, the different inhibitory properties of the organic acids influenced the required rate of secretion or rate of consumption (Fig. S3).

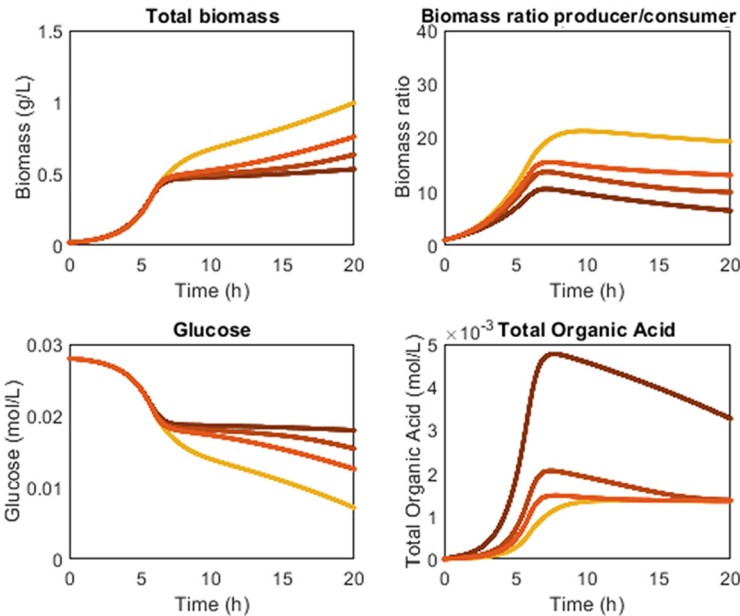

**FIG 10** Dynamic modeling of consortia interactions using push or pull mechanisms. The rate of organic acid production was varied from low (pull interaction, light orange lines) to high (push interaction, dark orange lines). Parameter values were selected based on the range of experimentally measured values. Push interactions reduced consortium performance including biomass accumulation and glucose conversion while pull interactions increased consortium performance by keeping acetic acid concentrations low. Models and parameters can be found in Fig. S3 and on github.

## DISCUSSION

Two synthetic *E. coli* consortia were constructed to test principles governing microbial interactions. The consortia were designed to catabolize glucose and unidirectionally exchange either acetic or lactic acid. Collectively, each consortium had the same genomic potential as wild-type *E. coli* K-12 since the genes deleted in one guild were present in the other guild. However, the two consortia displayed very different growth properties based on the partitioning of key genes. The LAE consortium, displaying a pull metabolite exchange motif, outperformed the WT based on four of six considered performance metrics, including the production of 55% more total biomass (mean = 0.48 $\pm$ 0.02 g cdw L$^{-1}$ and mean = 0.31 $\pm$ 0.01 g cdw L$^{-1}$, respectively; $n$ = 3; $P \ll 0.05$), the consumption of 86% more glucose (mean = 5.38 $\pm$ 0.8 mmol and mean = 2.89 $\pm$ 0.35 mmol, respectively; $n$ = 3; $P < 0.05$), and the production of 51% more biomass per H$^+$ secreted compared to the WT [mean = 2.16 $\pm$ 0.02 g (mol H$^+$)$^{-1}$ and 1.43 $\pm$ 0.06 g (mol H$^+$)$^{-1}$, respectively, $n$ = 3, $P \ll 0.05$; Fig. 7A to C]. Additionally, the LAE consortium consumed all reduced by-products by stationary phase, while the WT accumulated substantial acetic acid (Fig. 3, 5, and 7). These performance advantages came with the trade-off of a lower growth rate.

Synthetic consortia with defined components provide the basis for testing hypotheses and extracting design principles for community interactions. The first design principle central to the enhanced consortium properties observed in this study was the relationship between the relative rates of secretion and uptake of exchanged metabolites, e.g., push versus pull interaction motifs (56). The pull interaction motif used by the LAE consortium was sufficient to prevent substantial accumulation of the inhibitory metabolite, which created a benefit for both the consumer and producer. The balance of organic acid production and consumption in the LAE consortium modulated the environmental pH, which delayed the growth-arresting effects of accumulated organic acid and low pH experienced by the WT. A second consortial design rule gleaned from this study illustrated that no metabolite exchange is without cost for the producer; there are opportunity costs associated with uncaptured energy and often by-product inhibition (57) (Fig. 1 and 2). However, the opportunity cost for the producer guild can be off set through increased flux of glucose if consortial interactions can ameliorate the increase in organic acid secretion. The ProdL guild had a

higher opportunity cost [24 mol ATP (mol glucose)$^{-1}$] than the ProdA guild [22 mol ATP (mol glucose)$^{-1}$]. but the LAE consortium outperformed the AAE consortium due largely to the metabolite pull mechanism.

A third design principle demonstrates that environmental context such as buffering capacity dictates which metabolite exchange strategy, if any, will result in a competitive advantage for participants (58). Low buffering capacity is common in natural environments (29–32). The LAE consortium exhibited many enhanced performance metrics relative to the WT under low buffering conditions. Consistent with the thought experiment of the Darwinian Demon, which states it is not possible to optimize all fitness behaviors simultaneously (59), the LAE consortium had a reduced growth rate compared to the WT. Fast growth rates are major drivers of fitness in well-mixed, spatially homogenous laboratory environments where cultures are often supplied with abundant nutrients. In nutrient-poor environments, growth rates are reduced and the benefits of enhanced substrate depletion, enhanced biomass accumulation, and reduced H$^+$ production are likely important fitness metrics. The current study demonstrates that changing the buffering capacity changes the fitness of different phenotypes. The WT outperformed the LAE consortium when a conventional phosphate buffering capacity (64 versus 6.3 mM) was used (55). Enhanced buffering capacities also altered the consortia performance afforded by acetic acid exchange. A previous study measured a 15% enhancement in biomass productivity in an alternative acetic acid-exchanging *E. coli* consortium (46). The acetic acid producing strain in that study secreted approximately 10% of the glucose carbon as acetic acid compared to 30% to 50% in the current study. Tuning the secretion rate of the organic acid to match the consumption rate in the consumer is an important design parameter, as demonstrated in Fig. 10. These rates could be optimized in future studies using genetic engineering or adaptive evolution approaches.

Nutrient limitation is a common environmental challenge (60). Overflow phenotypes, where reduced by-products are secreted in the presence of external electron acceptors, can be competitive acclimations to nutrient limitation (51). As demonstrated in Fig. 2, abridged central metabolism pathways require fewer resources to construct but only partially oxidize substrates, highlighting the trade-off between metabolic pathway investment into proteins and the opportunity cost quantified by carbon source oxidation efficiency. WT *E. coli* secretes different organic acids, including acetic and lactic acids, in different quantities as a function of different nutrient limitations and degrees of nutrient stress (47). The secretion of reduced by-products in chemostats can lead to the evolution of cross-feeding populations with higher biomass titers (4) likely due to a combination of factors highlighted here, including more biomass per H$^+$ as well as the nonlinear relationship between enzyme velocity and the total required resource investment (enzymes and metabolites) to achieve that velocity (24, 45, 61, 62). This biomass yield benefit can potentially be realized in consortia without spatial segregation of populations (55).

The growth of each culture was arrested at a pH of 4.1 to 4.5 regardless of the initial pH. The pH decrease was primarily the result of two processes: organic acid secretion followed by H$^+$ dissociation and the consumption of ammonium for biomass synthesis, which released an H$^+$ when the nitrogen source was incorporated into biosynthetic molecules as ammonia. Thus, biomass production itself contributed to reaching the critical pH threshold. The WT drove the pH to the critical level via a combination of biomass production and acetic acid secretion. The LAE consortium, however, did not accumulate substantial organic by-products, allowing more biomass to be produced before reaching the critical pH threshold.

Exchange of organic acids in microbial consortia can also be analyzed from the perspective of other ecological principles that are typically applied at the macroscale. For example, the resource ratio theory suggests that two populations can cooperate to improve overall efficiency of resource usage. As an alternative to the competitive exclusion principle, it has been proposed that cooperating populations can exchange resources that they are each more efficient at utilizing and thereby more effectively deplete total resources (63). The two synthetic consortia effectively partitioned glucose and O$_2$ between two different populations. The ProdL guild was constructed by deleting the primary respiration chain enzymes including the O$_2$ cytochromes (50). ConsL required the external electron acceptor O$_2$ for the

complete oxidation of the organic acid. The lactic acid exchanged between the consortium members drove the glucose concentration lower than the WT generalist (Fig. 7); while not measured directly, it is proposed that the high biomass concentration in the LAE consortium drove the $O_2$ levels lower than the WT cultures. This performance would potentially classify the LAE consortium as a "super-competitor unit" as defined by de Mazancourt and Schwartz (63).

**Conclusion.** The presented work used synthetic consortia to test hypotheses governing microbial interactions mediated by push or pull metabolite exchange, quantified the inhibitory properties of the exchanged metabolites, calculated the opportunity costs associated with different phenotypes, and demonstrated the powerful role of environmental context on consortia performance. Ultimately, environment constrains whether division of labor strategies can enhance or decrease the fitness of participants.

## MATERIALS AND METHODS

**Culturing media.** Two different variations of M9 media were used. Conventional M9 medium for characterizing strain growth (64) contained 6 g $L^{-1}$ $Na_2HPO_4$ (42 mM), 3 g $L^{-1}$ $KH_2PO_4$ (22 mM), 1 g $L^{-1}$ $NH_4Cl$, and 0.5 g $L^{-1}$ NaCl. After autoclaving, 1 mL $L^{-1}$ 1 M $MgSO_4 \cdot 7H_2O$ solution was added along with 1 mL $L^{-1}$ trace metals solution, containing (per L): 0.73 g $CaCl_2 \cdot 2H_2O$, 0.10 g $MnCl_2 \cdot 4H_2O$, 0.17 g $ZnCl_2$, 0.043 g $CuCl_2 \cdot 2H_2O$, 0.06 g $CoCl_2 \cdot 6H_2O$, 0.06 g $Na_2MoO_4 \cdot 2H_2O$, and 0.24 g $FeCl_3 \cdot 6H_2O$. Carbon source (glucose, sodium acetate, or sodium lactate) was added to achieve the desired concentration from a filter sterilized stock solution. The pH of the medium was adjusted, if necessary, with HCl or NaOH, and the medium was then filter sterilized. For experiments investigating pH effect, conventional M9 medium was modified to ensure carbon limitation at 5 g $L^{-1}$ glucose by increasing nitrogen, iron, and sulfate content. Modified M9 medium contained 2.5 g $L^{-1}$ $NH_4Cl$, 1.5 mL $L^{-1}$ 1 M $MgSO_4 \cdot 7H_2O$, and an additional 2.4 mg $L^{-1}$ $FeCl_3 \cdot 6H_2O$. Low phosphate modified M9 medium contained 0.9 g $L^{-1}$ $Na_2HPO_4$ (6.3 mM) and no $KH_2PO_4$.

**Strains.** Mutant strains were derived from *E. coli* str. K-12 substr. MG1655. The wild-type strain served as the metabolic generalist for comparison with the mutant specialist strains.

**(i) Producer guild strains.** The lactate producer strain *E. coli* str. ECOM4LA was obtained from the Pålsson group (50). The strain was designed to prevent $O_2$ uptake through deletions of three terminal oxidases (*cbdAB*, *cydAB*, and *cyoABCD*), as well as deletion of the quinol monooxygenase *ygiN*; $O_2$ consumption was reduced ~60-fold and considered negligible. The strain was received in a frozen glycerol stock. Growth of the initial stock was slow, possibly due to freeze-thaw cycles. Thus, the stock was serially passaged by transferring during exponential growth phase to fresh conventional M9 media containing 4 g $L^{-1}$ glucose. Samples were tested for growth rate periodically, and serial passaging was continued until growth rate plateaued (~120 generations).

The acetate producer *E. coli* str. 409 was constructed from *E. coli* str. 307G100 ($\Delta aceA$, $\Delta ldhA$, $\Delta frdA$), which was designed in a previous study (46). This strain was acclimated to its altered genotype via chemostat growth for 100 generations. An additional deletion (*atpF*) was added to *E. coli* str. 307G100 using P1 viral transduction with the KEIO mutant library according to reference 39. *E. coli* str. 409 was designed to function similarly to the homoacetate producing strain reported in Causey et al. (48). The *atpF* gene deletion was confirmed with PCR (forward primer 5'-GTTATGGGTCTGGTGGATGC-3' and reverse primer 5'-CGAACACCAAAGTGTAGAACGC-3').

**(ii) Consumer guild strain.** The consumer strain *E. coli* str. 403 was previously constructed (46) to prevent glucose consumption by blocking the glucose phosphotransferase uptake system and phosphorylation of glucose via *ptsG*, *ptsM*, *glk*, and *gcd* deletions. This strain does not grow readily on glucose as the sole carbon source but is able to metabolize glucose at a slow base rate (0.016 $h^{-1}$), whereas lactate and acetate are readily consumed in the presence of $O_2$. The strain was acclimated to the gene deletions via growth in a chemostat for ~100 generations, as described previously (46).

**Batch culturing.** All cultures were grown at 37°C in a shaking incubator at 150 rpm. Frozen stock vials prepared from the same culture were used for all experiments. *E. coli* MG1655 and producer strain inocula were grown with 5 g $L^{-1}$ glucose as the carbon source, and consumer strain inocula were grown with either 1 g $L^{-1}$ sodium acetate or 2.8 g $L^{-1}$ sodium lactate. Disposable test tubes containing 5 mL modified M9, pH 7.0, were inoculated from frozen stock and incubated until optical density at 600 nm ($OD_{600}$) reached 0.2 to 0.4. Cultures were transferred into 25 mL modified M9, pH 7.0, in 250-mL borosilicate glass baffled shake flasks to an initial $OD_{600}$ ~0.010 and grown until $OD_{600}$ reached 0.2 to 0.4. Sufficient culture for inoculation was aliquoted into 15-mL Falcon tubes and pelleted at 4,000 rpm for 10 min at 20°C. Cells were washed in an equal volume of low phosphate modified M9 containing 5 g $L^{-1}$ glucose to remove metabolic by-products and excess phosphate and were pelleted again. One-hundred-milliliter experimental batch cultures of low phosphate-modified M9 containing 5 g $L^{-1}$ glucose were inoculated to an initial $OD_{600}$ ~0.020. Five-hundred-milliliter borosilicate glass baffled shake flasks with silicone sponge closures were used to allow gas exchange with minimal evaporation. Consortia were inoculated to an initial $OD_{600}$ ~0.020 of producer and consumer each (to provide the same density of primary glucose-consuming strain in all conditions).

Flasks were sampled aseptically approximately every doubling time; total culture volume was not reduced more than 20% by the end of the experiment. $OD_{600}$ and pH were measured, and culture supernatant was frozen at −20°C for subsequent metabolite analysis. For consortia experiments, samples were also analyzed for producer and consumer populations. Samples were serially diluted 1:10 in phosphate-buffered saline (PBS; 42.5 mg $L^{-1}$ $KH_2PO_4$ and 405.5 mg $L^{-1}$ $MgCl_2 \cdot 6H_2O$) and drop plated (10 10-$\mu$L drops) on selective conventional M9 agar after the method of reference 65. Selective plates contained 15 g $L^{-1}$ Noble agar (Affymetrix) to eliminate bacterial growth on carbon source contaminants available in the agar. The consumer was selected with either 1 g $L^{-1}$ sodium lactate or 1 g $L^{-1}$ sodium acetate, and the producer was selected with 1 g $L^{-1}$

glucose. In the lactate consortia experiments, both consumer and producer were found capable of growth on glucose agar plates due to the slower relative growth rate of the producer and the high yield of lactate. Therefore, the producer proportion of the population was estimated by subtracting the consumer counts on lactate agar from the total population (producer and consumer) counts on glucose agar. Experiments were performed with triplicate flasks for each condition along with an uninoculated control flask carried through the entire inoculation and sampling procedure. For all experimental measurements described, data were tested for normality using the Shapiro-Wilk test and significance between treatments was determined with Student's $t$ test (two sample $t$ test assuming unequal variance).

Experimental data used in the article are available in the supplemental data set available at https://github.com/rosspcarlson/becketal-syntheticconsortia.

**Dry weight and CFU correlations.** Correlation of $OD_{600}$ to cell dry weight (cdw) was determined for the producer and consumer strains separately. Producer strains were grown in conventional M9 containing 4 g $L^{-1}$ glucose, and the consumer strain was grown in conventional M9 containing 2.8 g $L^{-1}$ lactate. Exponentially growing cultures were harvested on ice, pelleted at 4,000 rpm for 20 min at 4°C in 50-mL Falcon tubes, resuspended in an equal volume of carbon-free M9 to minimize lysis of cells, and pelleted again. The cultures were combined and concentrated into one tube, and a series of 12 dilutions ranging from $OD_{600}$ 0.25 to 2.5 was made using carbon-free M9 as the diluent. $OD_{600}$ of each dilution was measured and recorded, and 5 mL of each dilution was aliquoted into a predried and preweighed aluminum pan. Three aluminum pans contained 5 mL carbon-free M9 as a media control. Pans were placed in a drying oven at 80°C for 24 h and weighed. Samples were dried for an additional 24 h and weighed again to ensure that samples were completely dry. Biomass concentration was calculated from the difference in mass. Correlation curves were constructed by setting the mass of the media control as zero and adjusting the sample masses accordingly. The cell dry weight correlation for *E. coli* MG1655 was obtained from reference 66.

Colony-forming unit (CFU) to $OD_{600}$ correlation curves were constructed in a similar manner. Exponentially growing cultures were harvested and a series of eight dilutions was made, ranging from $OD_{600}$ 0.010 to 0.275 for *E. coli* str. ECOM4LA and 0.010 to 1.1 for *E. coli* str. 403. $OD_{600}$ was measured and recorded, and each dilution was serially diluted 1:10 in PBS and drop plated on LB agar (65). Plates were incubated overnight, counts from appropriate dilutions (3 to 30 colonies within a drop) were recorded, and values (CFU $mL^{-1}$) were calculated.

**Metabolite analysis.** Glucose consumption and organic acid secretion were monitored via high-performance liquid chromatography (HPLC). An Agilent 1200 HPLC instrument was used with filtered (0.22 $\mu$m) 5 mM sulfuric acid as the mobile phase. Samples were stored in a chilled (4°C) autosampler during the run, and 20-$\mu$L sample injections were run on a Bio-Rad HPX-87H column operated at 45°C. Glucose was detected with refractive index detector with heater set at 40°C, and concentrations were normalized by fucose. Organic acids acetate and lactate were detected with variable wavelength detector. Cells were pelleted from samples and supernatant was frozen at −20°C until analysis, before which samples were filtered (0.45 $\mu$m) and prepared 1:1 (vol/vol) with 2× mobile phase containing fucose as an internal standard. Concentrations were quantified using calibration curves with limit of detection at 0.1 mM. Formate and succinate standards were also measured, but significant levels were not detected in the samples.

**Kinetic expression modeling.** A compendium of possible kinetic expressions to describe the inhibition effect of organic acids was identified (44) and examined against the lactate and acetate inhibition data both with and without glucose. Additionally, dual substrate models were explored to improve model fit for data with both glucose and lactate present in the medium (67), given the observation of co-consumption of the two compounds. Equations and parameters were defined in Python, and the data were fit using the Levenberg-Marquardt algorithm in the SciPy library (scipy.org). Goodness-of-fit was assessed visually, with the chi-square test function in SciPy, and via $R^2$ metrics. The Python fitting routines used are available at https://github.com/rosspcarlson/becketal-syntheticconsortia.git.

To model the dynamics of consortia growth compared to producer and generalist monocultures, a set of ordinary differential equations describing growth, substrate and product concentrations, and pH (based on acidification by organic acid by-products and using a yield to compensate for protons contributed by consumption of ammonium) was developed based on Monod kinetics. The ode45 solver in MATLAB was used to simulate batch growth. The model files and parameter set used are included in Fig. S3, and the MATLAB code is available at https://github.com/rosspcarlson/becketal-syntheticconsortia.

**Metabolic modeling.** A published *E. coli* central metabolism model was used to quantify the metabolic efficiencies of the guilds (51, 52). Distinct guild models were created by inactivating the appropriate reactions. The *in silico* model was decomposed into elementary flux modes using CellNetAnalyzer v.2018.1 (68), and individual elementary flux modes were identified using Excel and the noted optimization criteria. The *in silico* model can be found in Table S1 and S2.

## SUPPLEMENTAL MATERIAL

Supplemental material is available online only.

**FIG S1**, PDF file, 0.5 MB.
**FIG S2**, PDF file, 0.1 MB.
**FIG S3**, PDF file, 0.6 MB.
**TABLE S1**, XLSX file, 0.02 MB.
**TABLE S2**, XLSX file, 0.3 MB.
**TABLE S3**, XLSX file, 0.01 MB.
**TABLE S4**, XLSX file, 0.01 MB.

## ACKNOWLEDGMENTS

The study was supported by NSF awards DMS 1361240 and DGE 0654336 and NIH award U01EB019416. A.E.B. was supported in part by the USDA National Institute of Food and Agriculture (Hatch 1018813) and the M.J. Murdock Charitable Trust Biology Start-up Grant to Carroll College. Undergraduate researchers K.P., A.S., T.J., A.B., and M.D. were supported in part by the Montana State Undergraduate Scholars program and the National Institute of General Medical Sciences of the NIH under award no. P20GM103474.

The content is solely the responsibility of the authors and does not necessarily represent the official views of NIH.

We also thank the Pålsson research group for the gift of the ECOM4LA strain.

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
