## [Reviewer comments · mSystems]

Environment constrains fitness advantages of division of labor in microbial consortia engineered for metabolite push or pull interactions

Ross Carlson, Ashley Beck, Alissa Bleem, Jeffrey Heys, Tomas Gedeon, Hans Bernstein, Diana Schepens, William Harcombe, Kathryn Pintar, Ashley Schrammeck, Timothy Johnson, and Martina Du

Corresponding Author(s): Ross Carlson, Montana State University

Review Timeline:

Submission Date:	January 19, 2022
Editorial Decision:	March 2, 2022
Revision Received:	May 26, 2022
Accepted:	June 6, 2022

Editor: Alejandra Rodríguez-Verdugo

Reviewer(s): Disclosure of reviewer identity is with reference to reviewer comments included in decision letter(s). The following individuals involved in review of your submission have agreed to reveal their identity: Davide Ciccarese (Reviewer #2)

Transaction Report:

DOI: <https://doi.org/10.1128/msystems.00051-22>

March 2, 2022

Dr. Ross P Carlson
Montana State University
Chemical and Biological Engineering
Bozeman, MT 59717

Re: mSystems00051-22 (Environment constrains fitness advantages of division of labor in microbial consortia engineered for metabolite push or pull interactions)

Dear Dr. Ross P Carlson:

Thank you for submitting your manuscript to mSystems. We have completed our review and I am pleased to inform you that, in principle, we expect to accept it for publication in mSystems. However, acceptance will not be final until you have adequately addressed the reviewer comments.

The reviewers agreed that this is an important study that will advance the field of synthetic microbial ecology. The two reviewers have made excellent suggestions for improvement that should be addressed before we can accept your work for publication. The main point that is brought up and needs to be addressed is that some of the statistical analyses and model need refining. It was also noted that additional analyses and/or control experiments could be performed to strengthen some of the conclusions (reviewer #1). If this is not possible, please be explicit about the limitations of your data. Finally, even though the paper is clear and well written, two reviewers provided detailed comments on how to further improve the presentation and clarity of the manuscript. I look forward to receiving a revised manuscript.

Preparing Revision Guidelines

Sincerely,

Alejandra Rodríguez-Verdugo

Editor, mSystems

Journals Department
Reviewer comments:

Reviewer #1 (Comments for the Author):

The authors use synthetic communities to study the dynamics of consortia that cross-feed on organic acids. They compare two consortia, based on acetate and lactate cross-feeding, and show that they have a different interaction motif: for acetate, secretion by producers is faster than uptake by consumers and it thus accumulates in the growth medium (push motif); for lactate, uptake matches secretion, preventing accumulation in the media (pull motif). The authors use experiments to quantify the inhibitory effects of acetate and lactate, and metabolic modeling to quantify the opportunity costs involved with cross-feeding. Subsequently they compare the growth rate, yield, and biomass production of WT populations with the two cross-feeding consortia. The acetate cross-feeding community does worse than WT on all measures. In contrast the lactate cross-feeding community reaches higher yield and biomass than WT in weakly buffered media, but not in strongly buffered media. Based on this the authors argue that the pull motif prevents the accumulation of intermediates which inhibit cell growth in weakly buffered media. The cross-feeding interaction comes at the cost though to the maximum growth rate.

Overall the manuscript is well written, the methods and results are clearly explained, and most conclusions are supported by the presented data. Nonetheless, I have some suggestion for improvement.

1) The cross-feeding strains were constructed by deleting multiple genes. These deletions could have pleiotropic effects beyond the change in catabolic fluxes. For example, they could lead to metabolic bottlenecks, intercellular accumulation of inhibiting substances, deficiencies in anabolic pathways, or other pleiotropic effects that reduce growth and yield beyond the direct effects (catabolic flow and ATP yield) discussed by the authors.

The authors base some of their conclusions on comparisons of the fitness of the different consortia; I thus think is critical that they discuss to what extend these fitness differences could be caused by the kind of pleiotropic effects mentioned above. Can the authors rule out such effects based on previous work, their metabolic models, or control experiments?

2) The authors state that their model can capture the experimentally observed differences between low and high buffered media (line 332), however for the LAE consortia the model predictions (especially for biomass) match the data rather poorly, especially in the high buffered media. Also for the WT the predictions for the biomass show a substantial deviation. Unless the authors can improve the fit of their model (e.g. by optimizing the parameters) this statement needs to be toned down.

Moreover, I missed a detailed description of the full model; sheet 7 only shows figures + parameters but not the full equations, and I also could not find the Matlab code. Please add a description of the model (equations + initial conditions) to the SI (ideally as pdf) and upload code to repository.

3) The manuscript largely consists of two parts: the characterization of the inhibitory effects and opportunity costs (figures 1&2) and the measurement of the consortia growth (figures 3-8). At the moment these two parts feel a bit disconnected; they are pulled together in the discussion session, but before that the relation is not completely clear. I think it will be helpful for the reader if the different parts are integrated a bit more, by making more explicit early on in the text how the different parts of the manuscript are related (i.e. how the data shown in fi 1-2 can be used to understand the consortia dynamics).

One reason for the feeling of a disconnect is that the authors very carefully quantify the inhibition strength and opportunity costs, but it is not fully clear how these quantitative details are later used to interpret the consortia data. I guess that part of this data feeds into the model, but this is not made very explicit in the main text. I think making this explicit would help make the link more clear.

Also, adding a more detailed description of how the model was setup and parameterized (in text form with equations, and not as a table as in the current SI) would help make this point more clear.

Minor points:

4) line 92 "to test two consortia interaction hypotheses" & line 420 "to test hypotheses"

The authors do not really formulate clear, falsifiable, hypotheses in the intro, but rather a number of open questions. I would adjust phrasing here accordingly to make this clear.

5) line 118 "more inhibitory"

I found the phrasing "more inhibitory" in this section a bit confusing, and I would suggest using "lower inhibitory concentration" (or something similar) instead, to make clear that the main difference is in the K_i values of the compounds.

6) Table 1: There seems to be an error in the Ace+Glu equation, based on units it should be $\exp(-A / K)$ (not $K \cdot A$)

7) Table 1: I would use consistent notation throughout, to make comparisons easier. i.e. μ_{\max} and μ_C basically measure the same thing (max growth rate) and I would thus use the same symbol.

In the Ace+Glu model I would use K_I (instead of K) as above and likewise in the Lac+Glu model, I would replace α with $1/K_I = \alpha$ such that its value can directly be compared with the other K_i

8) Table 1: I'm surprised by the big difference in μ_{\max} and K_G between the Glu+Ace and Glu+Lac model. When $L=A=0$ these two models describe the exact same growth conditions, and both equations simplify to the same functional form, I would thus expect them to have the same value for μ_{\max} and K_G .

My guess is that the model has multiple parameter combinations that fit the data well. To make the other parameters easier to interpret, I suggest trying to force the μ_{\max} and K_G values to be the same for the Glu+Lac and Glu+Ace conditions (e.g. fit the two conditions simultaneously, or obtain these values from literature or an independent control experiment).

9) Table 2: Please comment on how the Glu/Lac/Ace concentrations were chosen. Are these the ones that maximize growth? Also, please add the concentrations in Molar units (in addition to the g/L) to make it easier to compare the values here to those shown in Table 1 and Figure 1.

10) Fig 6: These panels are very dense (at early time points) making it hard to see the data clearly.

I suggest making the symbols smaller or axis bigger (e.g. 1 column figure) to make it easier to see what happens at early times.

11) SI: The presentation of the SI is not optimal at the moment. I suggest the authors provide figures + legends (sheets 1,2 & 7) as a pdf file and upload the code (sheet 1) on Github (or similar).

The remaining sheets work well as Excel files.

Reviewer #2 (Comments for the Author):

The paper explores the division of labor as a response to nutrient availability comparing specialist consortia to generalist. Two types of interaction are reported. One is referred to as the 'push' interaction motif, where acetic acid is secreted faster by the producer than its consumption rate by the consumer. The other, 'pull' interaction, is where the consumer consumes lactic acid as quickly as the producer releases it.

The buffer condition of the experiment was set to investigate the effect on the fitness of the different specialist consortia or generalist. Thus, based on the author's view, these buffer conditions represent environmental constraints that eventually drive the interaction within the synthetic communities. Finally, each synthetic consortium of specialists is compared to generalists, showing that the environmental context favors one or the other metabolic style.

The research work represents a significant advancement in the field because it explores the effect of initial environmental constraints to determine synthetic communities' metabolic interaction dynamic and how these interactions change the environment. Furthermore, in this research work, the environmental context is considered a crucial factor in shaping the metabolic interactions, which is often unexplored in synthetic ecology. Besides the novelty of the research work, the authors created a complete work combining several approaches to fill the gap of knowledge and answer all their research questions.

At its current state, the manuscript needs a few essential modifications.

I have a few major general comments, and then I will report each specific modification needed, line by line.

- When predicted values are plotted against experimental data, it would be necessary to report the goodness of fit. I would suggest adding these values either in each plot or in a table in the supporting material.
- In several plots, the error bar is not visible. Reducing the size of each data point would make the error bar more visible. Please consider changing it.
- Throughout the text, the mean values, standard deviations, and the number of replicates should be reported. Each time, in the text, is indicated a significant differences between treatments; it should be tested with the appropriate statistical test and reported. Testing for homoscedasticity and normality should be performed to decide which statistical test to use.
- Changing the file format of the supplementary material would improve the readability (e.g. word instead of excel).
- Each time a percentage is reported, please also add at least the minimal information of mean values, standard deviation, and the number of replicates.
- Most of the information reported in the supplementary material is not comprehensively mentioned in the main text. For

instance, if the authors prefer the excel format, when supplementary material are cited in the main text it would be essential to report the sheet number in the main text.

- Overall, when referring to the environmental constrain, the initial buffer capacity are intertwined with the metabolite consumed. Therefore, specific metabolisms shape the pH differently. I believe that the introduction would benefit on elaborating more the role of the type of metabolites available in shaping the pH condition thus the environment.
- In Figures 3,4,5 several data points are missing. It would be helpful to read the reason for reporting different numbers in the data point.
- In figure 7, the spacing between different types of points makes it hard to distinguish them. It would be easier to separate them.
- In figure 8, giving more space between data points would help avoid overlap, and it'll increase readability.
- Please, report a reference for the conventional buffer capacity chosen in this study.

Line 140-141: The goodness of fit values should be reported either in each figure or in table 1.

145-147 In this sentence is stated that the published equation do not fit well the data. Therefore, it would be essential to report the goodness of fit and define criteria for discarding these equations.

Line 238-241: Please consider reporting mean value, standard deviation, number of replicates, and the significance of the difference between treatments.

Line 242-243: Please indicate the number of sheets of the excel file.

Line 245-250: Where is stated, 'highest' please consider to report, mean value, standard deviation, number of replicates and when necessary the significance of the difference between treatment when is indicated in the text.

Line 251: Besides the percentage (50%), please report the mean value, standard deviation, number of replicates and significance test and p-value.

Line 259: When indicating the producer to consumer ratio, please report the mean value of the ratio, standard deviation, number of replicates.

Line 261-264: Please, each time is stated 'highest' report mean value, standard deviation, number of replicates significance test, and p-value.

Line 277: When indicating the producer to consumer ratio, report mean value, standard deviation, number of replicates.

Line 301-306 The paragraph would benefit from reporting at the very beginning the number of the figure to which you are referring.

Line 311-316: Please, when comparing the synthetic consortium with the WT, report means value, standard deviation, number of replicates, and the appropriate significance test with the p-value.

Line 322-325: Please report mean value, standard deviation, number of replicates, and the appropriate significance test with the p-value.

Line 342-345: Please report together with the percentage report the mean value, standard deviation, number of replicates, and the appropriate significance test with the p-value.

Line 374: Please report at least a reference.

Line 382: Please report at least a reference.

Line 406: Please report at least a reference.

Line 408: Please report at least a reference.

Line 434-436, the increased nitrogen, iron, and sulfate concentration should be reported-additionally, the reasoning behind choosing these concentrations (e.g., to maintain a ratio of 1:5).

Responses to Reviewer Comments

Reviewer #1 (Comments for the Author):

The authors use synthetic communities to study the dynamics of consortia that cross-feed on organic acids.

They compare two consortia, based on acetate and lactate cross-feeding, and show that they have a different interaction motif: for acetate, secretion by producers is faster than uptake by consumers and it thus accumulates in the growth medium (push motif); for lactate, uptake matches secretion, preventing accumulation in the media (pull motif).

The authors use experiments to quantify the inhibitory effects of acetate and lactate, and metabolic modeling to quantify the opportunity costs involved with cross-feeding.

Subsequently they compare the growth rate, yield, and biomass production of WT populations with the two cross-feeding consortia.

The acetate cross-feeding community does worse than WT on all measures. In contrast the lactate cross-feeding community reaches higher yield and biomass than WT in weakly buffered media, but not in strongly buffered media. Based on this the authors argue that the pull motif prevents the accumulation of intermediates which inhibit cell growth in weakly buffered media. The cross-feeding interaction comes at the cost though to the maximum growth rate.

Overall the manuscript is well written, the methods and results are clearly explained, and most conclusions are supported by the presented data.

Nonetheless, I have some suggestion for improvement.

1) The cross-feeding strains were constructed by deleting multiple genes. These deletions could have pleiotropic effects beyond the change in catabolic fluxes. For example, they could lead to metabolic bottlenecks, intercellular accumulation of inhibiting substances, deficiencies in anabolic pathways, or other pleiotropic effects that reduce growth and yield beyond the direct effects (catabolic flow and ATP yield) discussed by the authors.

The authors base some of their conclusions on comparisons of the fitness of the different consortia; I thus think is critical that they discuss to what extend these fitness differences could be caused by the kind of pleiotropic effects mentioned above. Can the authors rule out such effects based on previous work, their metabolic models, or control experiments?

AUTHOR RESPONSE: Thank you for all your comments. We appreciate your time and expertise. Your review has strengthened the manuscript.

We agree there are often pleiotropic effects associated with gene deletions. The strains presented here have deletions in the central metabolism to modulate their phenotypes. The mutant strains likely have different tolerances to organic acids based on perturbations to central metabolism, altered proton motive force, redox state, ATP/ADP charge, etc. We have made three changes to address this point. First, we have added details to the materials and methods regarding strain development. The consumer and lactate overproducing strains were acclimated to the mutated genotypes prior to the presented experiments. The organic acid consumer was grown for 100 generations on acetate prior to consortia studies and the lactate producing strain was acclimated via

~120 generations prior to the consortia studies. The parent strain to the acetate overproducing strain was acclimated for 100 generations prior to the final gene deletion. This strain had a robust growth rate, so no further acclimation experiments were performed. Second, we have added additional data to the supplementary material demonstrating that the consumer strain has an increased sensitivity to acetic acid as compared to the wildtype. Thirdly, we have added text to the results mentioning the possibility of pleiotropic effects. However, we do not believe this change in organic acid sensitivity changes the interpretation of the interacting consortia. In fact, it strengthens the interpretations because the benefits of cross feeding compensate for an increased sensitivity to organic acids.

2) The authors state that their model can capture the experimentally observed differences between low and high buffered media (line 332), however for the LAE consortia the model predictions (especially for biomass) match the data rather poorly, especially in the high buffered media. Also for the WT the predictions for the biomass show a substantial deviation. Unless the authors can improve the fit of their model (e.g. by optimizing the parameters) this statement needs to be toned down.

Moreover, I missed a detailed description of the full model; sheet 7 only shows figures + parameters but not the full equations, and I also could not find the Matlab code. Please add a description of the model (equations + initial conditions) to the SI (ideally as pdf) and upload code to repository.

AUTHOR RESPONSE: Thank you for the comments and careful reading. The ODE model and results have been reanalyzed and the code has been uploaded to github. The ODE model is presented as a tool to further support the two hypotheses of the manuscript.

The inhibition equations used in the ODE model were measured with wildtype cells; as mentioned above, the consortia mutants likely have different sensitivities to organic acids. We made a conscience decision to avoid playing parameter games trying to fit the mutant strains to the equations. Instead, we demonstrate how relatively simple ODE models support the two proposed hypotheses. The ODE models show 1) the role of buffering environment on consortia and wildtype cultures (consortia produces more biomass at low buffer, but wildtype grows faster and produces more biomass at high buffer), and 2) the role of push vs pull interactions on consortia productivity consistent with the experimental data.

The text has been updated to report the key findings of the ODE model and to support the presented hypotheses. Additionally, two new figures have been added to the document to highlight the predictive and interpretative capabilities of the model.

3) The manuscript largely consists of two parts: the characterization of the inhibitory effects and opportunity costs (figures 1&2) and the measurement of the consortia growth (figures 3-8). At the moment these two parts feel a bit disconnected; they are pulled together in the discussion session, but before that the relation is not completely clear. I think it will be helpful for the reader if the different parts are integrated a bit more, by making more explicit early on in the text how

the different parts of the manuscript are related (i.e. how the data shown in fi 1-2 can be used to understand the consortia dynamics).

One reason for the feeling of a disconnect is that the authors very carefully quantify the inhibition strength and opportunity costs, but it is not fully clear how these quantitative details are later used to interpret the consortia data. I guess that part of this data feeds into the model, but this is not made very explicit in the main text. I think making this explicit would help make the link more clear.

Also, adding a more detailed description of how the model was setup and parameterized (in text form with equations, and not as a table as in the current SI) would help make this point more clear.

AUTHOR RESPONSE: Thank you for the comment, this is an excellent point. The ODE model serves to link the two parts of the study into a predictive and interpretative tool. As mentioned above, the results section has been expanded to highlight the strengths of the ODE model and 2 additional figures have been added to the manuscript. The parameters used in the model are listed in the supplementary material and are based on the experimental values. An explanation of the pH equation is provided for download on github.

Minor points:

4) line 92 "to test two consortia interaction hypotheses" & line 420 "to test hypotheses"
The authors do not really formulate clear, falsifiable, hypotheses in the intro, but rather a number of open questions.

I would adjust phrasing here accordingly to make this clear.

AUTHOR RESPONSE: Thank you for the comment. You are correct. We have reworded the hypotheses and now reference those two hypotheses throughout the manuscript.

5) line 118 "more inhibitory"

I found the phrasing "more inhibitory" in this section a bit confusing, and I would suggest using "lower inhibitory concentration" (or something similar) instead, to make clear that the main difference is in the K_i values of the compounds.

AUTHOR RESPONSE: Thank you for the suggestion. We have reworded the sentence to: 'Acetic acid was inhibitory to growth at lower total concentrations than lactic acid.'

6) Table 1: There seems to be an error in the Ace+Glu equation, based on units it should be $\exp(-A / K)$ (not $K \cdot A$)

AUTHOR RESPONSE: Thank you for the careful reading of the material. Considering your comments, we have reevaluated the inhibition equations and parameter fits. The revised

values are presented in Table 1 and in the supplementary material. Additionally, the code is available on github.

7) Table 1: I would use consistent notation throughout, to make comparisons easier. i.e. μ_{\max} and μ_C basically measure the same thing (max growth rate) and I would thus use the same symbol.

In the Ace+Glu model I would use K_I (instead of K) as above and likewise in the Lac+Glu model, I would replace α with $1/K_i = \alpha$ such that its value can directly be compared with the other K_i

AUTHOR RESPONSE: Thank you for the comments. We have reanalyzed the inhibition equations with the goal of reducing the number of free parameters and to make the notation more consistent. For example, the glucose half saturation constant (KG) was set based on literature so the parameter analysis would focus only on the organic acid inhibition parameters.

8) Table 1: I'm surprised by the big difference in μ_{\max} and K_G between the Glu+Ace and Glu+Lac model. When $L=A=0$ these two models describe the exact same growth conditions, and both equations simplify to the same functional form, I would thus expect them to have the same value for μ_{\max} and K_G .

My guess is that the model has multiple parameter combinations that fit the data well. To make the other parameters easier to interpret, I suggest trying to force the μ_{\max} and K_G values to be the same for the Glu+Lac and Glu+Ace conditions (e.g. fit the two conditions simultaneously, or obtain these values from literature or an independent control experiment).

AUTHOR RESPONSE: Thank you for the comment. You are correct, there are many alternative solutions. We have reanalyzed all the parameter fits with a focus on only the inhibition parameters. This was done by setting the experimentally measured μ_{\max} values and the published glucose half saturation value. We have also added R^2 values to all inhibition expressions to facilitate analysis.

9) Table 2: Please comment on how the Glu/Lac/Ace concentrations were chosen. Are these the ones that maximize growth?

Also, please add the concentrations in Molar units (in addition to the g/L) to make it easier to compare the values here to those shown in Table 1 and Figure 1.

AUTHOR RESPONSE: Thank you for the comment. The table caption has been modified to include the mM concentrations in addition to the $g\ L^{-1}$ concentrations. The organic acid concentrations were picked to be near the maximum growth rate based on the total organic acid concentration.

10) Fig 6: These panels are very dense (at early time points) making it hard to see the data clearly.

I suggest making the symbols smaller or axis bigger (e.g. 1 column figure) to make it easier to see what happens at early times.

AUTHOR RESPONSE: Thank you for the suggestion. We have made the points smaller to facilitate the viewing of the data.

11) SI: The presentation of the SI is not optimal at the moment. I suggest the authors provide figures + legends (sheets 1,2 & 7) as a pdf file and upload the code (sheet 1) on Gitbub (or similar).

The remaining sheets work well as Excel files.

AUTHOR RESPONSE: Thank you for the suggestion. We have reworked the supplementary material. Each item is uploaded as a separate pdf or excel file. The Python code for the inhibition equations and Matlab ODE model have also been uploaded to github.

Reviewer #2 (Comments for the Author):

The paper explores the division of labor as a response to nutrient availability comparing specialists consortia to generalist. Two types of interaction are reported. One is referred to as the 'push' interaction motif, where acetic acid is secreted faster by the producer than its consumption rate by the consumer. The other, 'pull' interaction, is where the consumer consumes lactic acid as quickly as the producer releases it.

The buffer condition of the experiment was set to investigate the effect on the fitness of the different specialist consortia or generalist. Thus, based on the author's view, these buffer conditions represent environmental constraints that eventually drive the interaction within the synthetic communities. Finally, each synthetic consortium of specialists is compared to generalists, showing that the environmental context favors one or the other metabolic style. The research work represents a significant advancement in the field because it explores the effect of initial environmental constraints to determine synthetic communities' metabolic interaction dynamic and how these interactions change the environment. Furthermore, in this research work, the environmental context is considered a crucial factor in shaping the metabolic interactions, which is often unexplored in synthetic ecology. Besides the novelty of the research work, the authors created a complete work combining several approaches to fill the gap of knowledge and answer all their research questions.

At its current state, the manuscript needs a few essential modifications.

I have a few major general comments, and then I will report each specific modification needed, line by line.

- When predicted values are plotted against experimental data, it would be necessary to report

the goodness of fit. I would suggest adding these values either in each plot or in a table in the supporting material.

AUTHOR RESPONSE: First, we would like to thank you for your comments, we appreciate your investment of time and expertise. Your careful reading of the document and eye for detail has enabled us to improve the manuscript.

We have added statistical metrics for all fits.

- In several plots, the error bar is not visible. Reducing the size of each data point would make the error bar more visible. Please consider changing it.

AUTHOR RESPONSE: Thank you for the comment; we have remade many of the figures using smaller point sizes to facilitate readability.

- Throughout the text, the mean values, standard deviations, and the number of replicates should be reported. Each time, in the text, is indicated a significant differences between treatments; it should be tested with the appropriate statistical test and reported. Testing for homoscedasticity and normality should be performed to decide which statistical test to use.

AUTHOR RESPONSE: Thank you for the comment; for each of the specific values pointed out, we have added parenthetical notations with the mean value referred to, associated standard deviation, number of replicates, and Student's t-test p-value to support any statement of significant difference (lines 235-8, 244-6, 250-2, 260, 263-5, 279, 311-9, 325-9, 348-52). We first tested that a normal distribution is a reasonable assumption for the data using the Shapiro-Wilk test, thus verifying Student's t-test as an appropriate test for significance (these statistical analysis details have been added to the methods section as well in lines 500-2). We have also included in the text a special callout to Sheet 5 in the supplemental material which contains a table organizing all the means and standard deviation for the measurements reported (line 244).

- Changing the file format of the supplementary material would improve the readability (e.g. word instead of excel).

AUTHOR RESPONSE: Thank you for the comment. The supplementary material was been reformatted. Each item is a separate file while the python code for fitting the inhibition data and the Matlab ODE model have been uploaded to github.

- Each time a percentage is reported, please also add at least the minimal information of mean values, standard deviation, and the number of replicates.

AUTHOR RESPONSE: Thank you for the comment; for each of the specific percentages listed in the results section, we have added parenthetical notations with the mean value referred to, associated standard deviation, number of replicates, and Student's t-test p-

value to support any statement of significant difference (lines 250-2, 311-9, 325-9, 348-52).

- Most of the information reported in the supplementary material is not comprehensively mentioned in the main text. For instance, if the authors prefer the excel format, when supplementary material are cited in the main text it would be essential to report the sheet number in the main text.

AUTHOR RESPONSE: Thank you for the comment. We have added supplementary material file number designators.

- Overall, when referring to the environmental constrain, the initial buffer capacity are intertwined with the metabolite consumed. Therefore, specific metabolisms shape the pH differently. I believe that the introduction would benefit on elaborating more the role of the type of metabolites available in shaping the pH condition thus the environment.

AUTHOR RESPONSE: Thank you for the excellent suggestion. We have modified the introduction to references to two game theory studies.

- In Figures 3,4,5 several data points are missing. It would be helpful to read the reason for reporting different numbers in the data point.

AUTHOR RESPONSE: Thank you for the comment; subplots showing metabolite measurements have fewer data points than subplots showing biomass or pH measurements because metabolite samples were not taken as frequently as biomass or pH measurements to avoid excessively depleting culture volume over the length of the experiment. We have added a note to explain this reason in each of the captions for figures 3, 4, and 5 (lines 755-7, 763-4, 771-3).

- In figure 7, the spacing between different types of points makes it hard to distinguish them. It would be easier to separate them.

AUTHOR RESPONSE: Thank you for the comment. The format of Figure 7 and 8 has been changed to increase readability.

- In figure 8, giving more space between data points would help avoid overlap, and it'll increase readability.

AUTHOR RESPONSE: Thank you for the comment. The format of Figure 8 has been changed to increase readability.

- Please, report a reference for the conventional buffer capacity chosen in this study.

AUTHOR RESPONSE: Thank you for the comment; the conventional buffer capacity is specified by the original M9 medium recipe cited in the materials and methods (Ausubel et al., 1992). We have also added this reference to the results for clarity (line 324).

Line 140-141: The goodness of fit values should be reported either in each figure or in table 1.

AUTHOR RESPONSE: Thank you for the attention to detail; we have included R² values quantifying the statistics of each model.

145-147 In this sentence is stated that the published equation do not fit well the data. Therefore, it would be essential to report the goodness of fit and define criteria for discarding these equations.

AUTHOR RESPONSE: Thank you for the comment. This section has been reworded and each fit of experimental data now included an R² metric.

Line 238-241: Please consider reporting mean value, standard deviation, number of replicates, and the significance of the difference between treatments.

AUTHOR RESPONSE: Thank you for the suggestion; we have added mean value, standard deviation, and number of replicates for each measurement (lines 235-8). While our intention here was to simply report the highest and lowest values for the generalist across the tested pH conditions (descriptive statistics - min, max) and not necessarily to imply a statement of significance, we did also add a two-sample t-test result ($p \ll 0.05$).

Line 242-243: Please indicate the number of sheets of the excel file.

AUTHOR RESPONSE: Thank you for the comment. All references to supplementary material are now specific to individual files.

Line 245-250: Where is stated, 'highest' please consider to report, mean value, standard deviation, number of replicates and when necessary the significance of the difference between treatment when is indicated in the text.

AUTHOR RESPONSE: Thank you for the attention to detail; we have added mean value, standard deviation, and number of replicates for each measurement.

Line 251: Besides the percentage (50%), please report the mean value, standard deviation, number of replicates and significance test and p-value.

AUTHOR RESPONSE: Thank you for the attention to detail; we have added mean value, standard deviation, and number of replicates for each measurement, along with two-sample t-tests comparing the significance ($p < 0.05$) between the acetate consortium and producer.

Line 259: When indicating the producer to consumer ratio, please report the mean value of the ratio, standard deviation, number of replicates.

AUTHOR RESPONSE: Thank you for suggestion; we have added mean value, standard deviation, and number of replicates for the producer to consumer ratios.

Line 261-264: Please, each time is stated 'highest' report mean value, standard deviation, number of replicates significance test, and p-value.

AUTHOR RESPONSE: Thank you for the attention to detail; we have added mean value, standard deviation, and number of replicates for each measurement. While our intention here was to simply report the highest values for the lactate consortium across the tested pH conditions (descriptive statistics - max) and not necessarily to imply a statement of significance, we did also add a two-sample t-test result showing no significant difference between pH 6.5 and 7.0 conditions ($p > 0.05$).

Line 277: When indicating the producer to consumer ratio, report mean value, standard deviation, number of replicates.

AUTHOR RESPONSE: Thank you for the attention to detail; we have added mean value, standard deviation, and number of replicates for the producer to consumer ratios.

Line 301-306 The paragraph would benefit from reporting at the very beginning the number of the figure to which you are referring.

AUTHOR RESPONSE: Thank you for the comment; we have added a reference to Figure 7.

Line 311-316: Please, when comparing the synthetic consortium with the WT, report means value, standard deviation, number of replicates, and the appropriate significance test with the p-value.

AUTHOR RESPONSE: Thank you for the attention to detail; we have added mean value, standard deviation, and number of replicates for each percentage, along with two-sample t-tests for each comparison ($p << 0.05$) between the lactate consortium and generalist.

Line 322-325: Please report mean value, standard deviation, number of replicates, and the appropriate significance test with the p-value.

AUTHOR RESPONSE: Thank you for the attention to detail; we have added mean value, standard deviation, and number of replicates for each percentage, along with two-sample t-tests for each of the comparisons ($p < 0.05$) between the generalist and lactate consortium.

Line 342-345: Please report together with the percentage report the mean value, standard deviation, number of replicates, and the appropriate significance test with the p-value.

AUTHOR RESPONSE: Thank you for the attention to detail; we have added mean value, standard deviation, and number of replicates for each percentage, along with two-sample t-tests for each comparison ($p << 0.05$) between the lactate consortium and generalist.

Line 374: Please report at least a reference.

AUTHOR RESPONSE: Thank you for the comment. We have updated the ODE model which quantifies the potential role between different organic acid secretion rates and organic acid consumption rates. We now reference this figure when making the statement.

Line 382: Please report at least a reference.

AUTHOR RESPONSE: Thank you for the comment. The sentence in line 382: “Wild type *E. coli* secretes different organic acids, including acetic and lactic acids, in different quantities as a function of different nutrient limitations and different degrees of nutrient stress” already has a reference included at the end (reference #47). This reference is our own work and quantifies overflow metabolisms as a function of nutrient limitation

Line 406: Please report at least a reference.

AUTHOR RESPONSE: Thank you for the comment. We have included a reference to Figure 7 to support and clarify this statement.

Line 408: Please report at least a reference.

AUTHOR RESPONSE: Thank you for the comment. The sentence in line 382: “while not measured directly, it is proposed that the high biomass concentration in the LAE consortium drove the O₂ levels lower than the WT cultures” is a hypothesis statement to provide a potential explanation for the results we observed; thus, there is no direct reference to add in this sentence. We do include a reference in the sentence following (reference #63) to provide support and explanation from the literature.

Line 434-436, the increased nitrogen, iron, and sulfate concentration should be reported- additionally, the reasoning behind choosing these concentrations (e.g., to maintain a ratio of 1:5).

AUTHOR RESPONSE: The increased nitrogen, iron and sulfate were designed based on the elemental composition of wild type *E. coli* and to ensure the medium was carbon limited. The analysis was based on our previous analysis of carbon, nitrogen, or iron limited *E. coli* growth (reference 47)

June 6, 2022

Dr. Ross P Carlson
Montana State University
Chemical and Biological Engineering
Bozeman, MT 59717

Re: mSystems00051-22R1 (Environment constrains fitness advantages of division of labor in microbial consortia engineered for metabolite push or pull interactions)

Dear Dr. Ross P Carlson:

Thank you for addressing the reviewers' comments. I agree with your assessment that the quality of the documents has improved. That said, please be advised that the figures from this current version have low resolution and will need to be improved for publication. Looking forward to seeing your published work.

Your manuscript has been accepted, and I am forwarding it to the ASM Journals Department for publication. For your reference, ASM Journals' address is given below. Before it can be scheduled for publication, your manuscript will be checked by the mSystems production staff to make sure that all elements meet the technical requirements for publication. They will contact you if anything needs to be revised before copyediting and production can begin. Otherwise, you will be notified when your proofs are ready to be viewed.

Publication Fees:

We recognize that the video files can become quite large, and so to avoid quality loss ASM suggests sending the video file via <https://www.wetransfer.com/>. When you have a final version of the video and the still ready to share, please send it to mSystems staff at mSystems@asmusa.org.

For mSystems research articles, if you would like to submit an image for consideration as the Featured Image for an issue, please contact mSystems staff at mSystems@asmusa.org.

Sincerely,

Alejandra Rodríguez-Verdugo
Editor, mSystems

Journals Department
Table S4: Accept

Table S3: Accept

Figure S3: Accept

Table S1: Accept

Table S2: Accept

Supplemental Material: Accept

Supplemental Material: Accept